# The Influence of Surface Texturing of Ceramic and Superhard Cutting Tools on the Machining Process—A Review

**DOI:** 10.3390/ma15196945

**Published:** 2022-10-06

**Authors:** Sergey N. Grigoriev, Thet Naing Soe, Khaled Hamdy, Yuri Pristinskiy, Alexander Malakhinsky, Islamutdin Makhadilov, Vadim Romanov, Ekaterina Kuznetsova, Pavel Podrabinnik, Alexandra Yu. Kurmysheva, Anton Smirnov, Nestor Washington Solís Pinargote

**Affiliations:** 1Laboratory of Electric Current Assisted Sintering Technologies, Moscow State University of Technology “STANKIN”, Vadkovsky per. 1, 127055 Moscow, Russia; 2Department of High-Efficiency Machining Technologies, Moscow State University of Technology “STANKIN”, Vadkovsky per. 1, 127055 Moscow, Russia

**Keywords:** ceramic cutting tool, ceramic cutting materials, superhard cutting tool, superhard cutting materials, ultrahard cutting materials, surface texturing, textured cutting tools

## Abstract

Machining is an indispensable manufacturing process for a wide range of engineering materials, such as metals, ceramics, and composite materials, in which the tool wear is a serious problem, which affects not only the costs and productivity but also the quality of the machined components. Thus, the modification of the cutting tool surface by application of textures on their surfaces is proposed as a very promising method for improving tool life. Surface texturing is a relatively new surface engineering technology, where microscale or nanoscale surface textures are generated on the cutting tool through a variety of techniques in order to improve tribological properties of cutting tool surfaces by reducing the coefficient of friction and increasing wear resistance. In this paper, the studies carried out to date on the texturing of ceramic and superhard cutting tools have been reviewed. Furthermore, the most common methods for creating textures on the surfaces of different materials have been summarized. Moreover, the parameters that are generally used in surface texturing, which should be indicated in all future studies of textured cutting tools in order to have a better understanding of its effects in the cutting process, are described. In addition, this paper proposes a way in which to classify the texture surfaces used in the cutting tools according to their geometric parameters. This paper highlights the effect of ceramic and superhard textured cutting tools in improving the machining performance of difficult-to-cut materials, such as coefficient of friction, tool wear, cutting forces, cutting temperature, and machined workpiece roughness. Finally, a conclusion of the analyzed papers is given.

## 1. Introduction

As is known, machining is an indispensable manufacturing process for an extensive range of engineering materials, such as metals, ceramics, and composite materials. The development of new alloys for their application in the industry is related to the creation of new cutting materials in order to make their machinability a productive process [1]. Unfortunately, tool wear, which takes place during machining, is a serious problem, which affects not only the costs and productivity but also the quality of the machined components [2]. The common types of tool wear are crater and flank wear, which take place on both the rake face and flank face of a cutting tool, respectively. The most common causes of tool wear are abrasive wear, adhesive wear, diffusion wear, chemical wear, and fracture wear. In addition, friction at the tool–chip and tool–workpiece interfaces, which is associated with tool wear, is another problem that takes place during machining as it directly affects the cutting force and cutting temperature and also influences the generation of vibration in the system, which in turn further increases tool wear [3].

Different approaches are generally used to increase the tool life and their productivity during machining. For instance, a very important approach takes place in the early stages of the cutting tool creation, and it includes the correct design of the geometry and the choice of the cutting material. These two stages have a direct influence on the tool cost, its quality, and, likewise, the useful tool life. The development of new advanced tool materials, as well as optimization of processing parameters, are other ways to extend tool life [4,5,6]. Another approach is the use of high-performance cutting fluid during machining, which increases lubricity and cooling in the contact area, improving tool life [7]. Finally, a further way to increase tool life is the modification of the cutting tool surface, such as coating the cutting tool with a material that has high hardness properties [8,9,10]. Thus, the wide use of coatings such as TiN, TiCN, TiAlN, Al_2_O_3_ and many others are examples of this approach [11,12].

Although there are different approaches to improving tool life, most of them are expensive, complex and not environmentally friendly. For this reason, the need to find more appropriate methods to improve tool life is urgent. A very promising method for improving tool life is the modification of the cutting tool surface by the application of textures on their surfaces. Surface texturing is a relatively new method of creating micro- or nanoscale textures on the cutting tool surface through different techniques in order to improve the tribological properties of cutting tool surfaces. This improvement is achieved because of the simultaneous reduction in the coefficient of friction and increase in wear resistance [13]. This occurs because the textures on the surface act as reservoirs of lubricating oil that can be supplied when the amount of lubricant is minimal or absent [14]. Thus, the goal of using textures on cutting tools is to increase their productivity during machining.

In the literature, there have been published works on texturing the surface of cutting tools, in which their effect on friction and wear were studied [15,16,17,18,19,20,21,22,23,24]. These studies showed that this approach allows for improving the tribological properties in the tool–chip interface, reducing the machining temperature, the rate of tool wear and the cutting forces, and consequently obtaining an increase in machining productivity [25]. Additionally, due to the application of textures, a reduction in the temperature stress of the machining process, as well as the production cost by reducing the need for cutting fluid, is obtained [26]. These studies were mainly performed on cutting materials such as high-speed steel (HSS) and carbide tools due to their ease of surface texturing. On the other hand, ceramic and superhard cutting tools have also been studied, but on a smaller scale. This low number of publications is mainly related to the machining difficulty presented by ceramic and superhard cutting tools since the most common method to form textures was electrical discharge machining (EDM), which is applicable only for conductive materials.

The rapid development of technologies and the current facility of implementing powerful lasers makes it possible to generate textures with different characteristics and forms in materials that are difficult to work with, which offers many possibilities for future and more detailed studies in this area. Although there are different reviews focused on the texturing of cutting tools, they contain information that more than 95–99% are related to carbide tools, and there is no review that presents the works performed on texturing of ceramics and superhard cutting tool materials.

In this paper, the studies carried out to date on the texturing of ceramic and superhard cutting tools have been reviewed. For a better understanding of the subject, this manuscript has been divided into five sections. The first is the introduction. The second section summarizes the most common methods for creating textures on the surfaces of different materials. The third section describes the most common forms of textures used in cutting tools. Here, an attempt is made to classify and generalize the description of textures according to their geometric parameters. In the fourth section, the works carried out based on ceramics and superhard materials are cited, and the results obtained in these works are summarized based on the texture influence on tool wear, friction coefficient, force and cutting temperature. The last section is the conclusion of the analyzed papers. Table 1 shows the content of this paper.

## 2. Methods of Creating Texturing

The formation of different shapes, textures and structures on cutting tool materials is a difficult task due to their high hardness and complex geometry [27]. This is made even more difficult since creating textures is an accurate and exact process, which ensures the required precision and shape of the textures. Therefore, when choosing the appropriate machining process, important factors must be considered, such as the shape of the texture elements, their geometric dimensions, and the mechanical, electrical, and thermal properties of the material to be machined.

In this way, several common machining methods are used to create textures, such as laser machining (LM) (13) [28,29], laser beam [30], electrical discharge machining [31], wire electric discharge machining [32,33], grinding wheel textures [20], and plasma machining [34], as well as the new uncommon methods that are proposed more frequently in recent years. Some examples of these widely used machining methods are shown in Table A2, see Appendix A. From this table, it can be concluded that the research trend tends toward surface texturing by laser and electric discharge machining due to its flexibility and ability to form complex textures with microscale dimensions.

Traditional machining methods, such as grinding, can produce precise, high-quality microgrooves and they do not produce harmful gases and are environmentally friendly. However, the created microgrooves are formed one by one and, as a result, the process itself is inefficient. These methods are suitable for creating microgroove-based textures on turning and milling inserts [20].

The use of Rockwell-C and Vickers indentation [35] is another interesting method used to create textures; however, these techniques are not suitable when creating an array of dimples or when other shapes are required, as they are slow processes and do not provide a constant and precise pitch between the texture elements to be created. Common techniques for creating textures will be discussed in the following subsections.

### 2.1. Plasma Arc Machining Textures

Plasma arc machining can be used for the textured formation due to its possibility to reach temperatures as high as 33,300 °C. As a result of the greater efficiency of plasma arc systems, they are often used for machining metals. Moreover, in this process, there is no chemical reaction in the plasma arc between the gas and specimen. Unfortunately, the double arcing between the nozzles and the workpiece has a negative effect and can cause damage to both the electrode and the workpiece. In addition, high heat transfer rates, which can change the material structure, are found to place during plasma arc [36].

Sawant et al. [34] found that the power of the micro-plasma around 264 W and exposure time around 45 s were the best parameters for producing dimple texture with a high aspect ratio and almost circular shape on an HSS grade T-42. The authors concluded that they are optimum conditions for machining an array of dimple textures on the cutting tools. Moreover, plasma arc machining proved to be an economically and environment friendly method for machining dimple and spot textures.

### 2.2. Laser Surface Texturing

Laser surface texturing (LST) or laser machining is an effective technology for creating textures on any surface. Surface texturing created by femtosecond laser has a significant role in enhancing the coefficient of friction and wear rate when those are compared with untextured surfaces [29]. Because of the flexibility of this technology, the texture size, shapes, depth and density can be controlled easily by the laser parameters [37]. On the other hand, the laser process is restricted by defects of low quality, for instance, irregular edges of the micro-channels and irregular protrusions inside the textures [38].

Laser machining is usually used for machining different textures such as dimples [13,14,35,39], elliptical [28], linear [30,31,32,39,40], areal [31,40]. For instance, Sedlaček et al. [35] produced dimple texture by LST and compared it with an indented pyramid shape texture on P/M cold work steel. The dimple texture showed the lowest friction. This improvement of results has been observed even in cemented carbide cutting tools [39], where the dimple shape texture was compared with linear shape textures.

In the same way, Wang et al. [29] observed that microgrooves created by LST and with an inclination of 90° and 45° provide the minimum wear rate and friction coefficient in tribology testing performed on AISI 304L stainless steel. This could be explained by the contributions of the effects on friction with the variation of microgroove spacing under starved oil lubricated conditions. Moreover, the inclination angle between the sliding direction and the channel direction has a clear influence on the wear rate of the material.

Figure 1 shows that the energy of the laser pulse of micro pores (dimples) is about 5.7 mJ saturation value. The crater diameter stays approximately the same in the case of higher pulse energies even if the additional energy offered by the material does not induce a greater diameter. However, pore depth is reduced when the energy is increased with a minimum of 2.5 µm reached at an energy of about 5.7 mJ and maintained till 8.3 mJ [37].

One of the best advantages of using LST is the ability to machine any hard-to-cut materials such as ceramics, polymers and metals. Laser is a precision process and accurate, hence it gives the required shape and size. In the case of dimple formation, sublimation, melting, and vaporization are the procedures of the machining process. However, because of the laser beam pulse’s high intensity, the surface area in close proximity to the dimple also undergoes melting and rapid re-solidification as shown in Figure 2. That defect could not be cured, and high-power laser surface processing cannot solve that problem because of unacceptable surface roughness [37].

### 2.3. Electrical Discharge Machining

Electrical discharge machining (EDM) is one of the non-traditional machining processes and innovative methods, which remove materials thermally by sparks [41]. It is a process for machining electrically conductive materials in which the workpiece material is removed by heat generated by an electric current between the electrode and workpiece in the form of sparks in the presence of a dielectric fluid [42].

The EDM tool is featured with simplicity and is electrically conductive irrespective of the hardness of the tool. Complex shapes are accomplished easily without restrictions when machining difficult-to-cut materials such as super alloys and carbides [43,44]. Electrical discharge machining is used for machining external shapes, cavities, holes or profiles in an electrically conductive workpiece material. Machining occurs by melting or vaporizing the material in a certain area by controlled application of high-frequency electrical discharges [45]. Thus, EDM can create different texture shapes accurately and simply, hence the different texture shapes such as dimples and channels with circular, conical, triangle and D-shapes could be created by the EDM method [46]. The machined shape in the produced workpiece from the EDM process has an identical replica of the electrode shape [47], the wire electrical discharge machining is used for forming the channels with different inclination angles and the sinking electrical discharge machining is used for forming dimples with different sizes.

No cutting forces in EDM, mainly due to no direct contact between the electrode and workpiece, even more importantly complex shapes can be accomplished by the process. The versatile advantages of EDM such as machining three-dimensional complex shapes and hard-to-cut materials irrespective of their hardness [48], thanks to the removing material mechanism nature that depends on the thermal method. On the other hand, the productivity of EDM is relatively low owing to the natural laws of electrical erosion [49]. According to Hofy [36], the suitable parameters in order to accomplish the electrical discharge machining process are viewed as follows: the magnitude of voltage varies from 20 to 120 V, the frequency is about 5 Hz, and the gap between two electrodes is from 0.01 to 0.5 mm. The shape of pulse current and voltage were viewed in Figure 3 in the case of using an RC generator [50]. The gap between electrode and workpiece exists to support a place for occurring sparks, as a consequence for that, the material is removed and the frequency range of sparks is from 2000 to 50,000 sparks per second [42]. It is possible for the plasma temperature to reach 20,000 °C [51].

### 2.4. Focused Ion Beam Machining

Focused Ion Beam (FIB) machining is of great importance in the production process of micro-structured surfaces. The FIB milling process is based on the formation of patterns by the direct impact of the ion beam on the surface of the substrate material. In this process, the ions penetrate the material to be machined and they lose their energy while removing the atoms from the workpiece surface. This method has the advantage of making micro- and nanostructures with a resolution of up to 100 nm without using expensive and complicated masks and pattern transfer procedures.

There are different studies related to the application of FIB in the texturing process. For instance, Chang et al. [52] utilized FIB with liquid gallium ion sources in order to form microgrooves on a milling tool. The application of these microgrooves showed that the wear resistance of the milling cutting tool was improved. On the other hand, FIB milling was used by Nakano et al. [53] in order to study the effect of texturing on friction reduction. In their work, the authors formed micro-dimples with a diameter of 30 µm and a depth of 11 µm.

Some advantages of the FIB method are: high resolution regardless of the type of material to be machined, a lower roughness of the machined surface [54], and the use of an ion beam with a small diameter that provides extremely small lateral dimensions to the texture. The disadvantage of FIB is its processing speed, which is very low. Therefore, FIB is applicable for the texturing of areas with small dimensions.

### 2.5. Micro Grinding

Micro grinding is used to modify the surface of different materials in a simple and economical way [55]. This method is commonly used for microgrooving and micro-dimple formation because the grains in the wheels can remove small volumes of workpiece material. Despite this, the aforementioned methods used in micromachining always outperform micro grinding. The technical problem that takes place in micro grinding is the correct choice of the matrix as well as the type and size of the abrasive material for the wheels. This is because when trying to produce very small, and accurate grinding tool shapes the grain size of the abrasive material must be carefully considered [56]. This machining method can be used for the fabrication of textured surfaces with microgrooves. Lately, it has been possible to produce wheels with a thickness of the order of tens of micrometers, which facilitate the manufacture of narrow grooves in the order of this size. This allows the micro grinding technique to be used for the production of 2D/3D microgrooves [57]. For example, Xie et al. [58] demonstrated that textured tools obtained by micro grinding can reduce tool wear in dry cutting.

### 2.6. Conclusions of Section 2

Texturing is an effective technique to increase the tribological properties of cutting tools without the need to resort to classical methods such as changing the chemical composition and geometry in order to improve their performance. Therefore, knowledge and understanding of the methods currently used for texturing are necessary.

There are many methods to create textured shapes, however, based on the survey of more than 150 papers, it is found that the most common methods for texturing could be summed up as Laser Surface Texturing (LST), Electrical Discharge Machining (EDM), Focused Ion Beam (FIB), micro grinding, plasma machining and texturing from hardness devices. The technique, the scientific principle, the basic concepts, the advantages, the disadvantages, the materials of the products and the results were seen in detail in this section.

These methods could be classified into three categories as follows:Non-traditional machining methods (LST, EDM, FIB and plasma machining, which are the most common for texturing).Traditional machining methods (micro grinding and others that are less common for texturing).Textures created by hardening devices (a technique that uses hardness devices for plastic deformation of the surface of materials to create textures).

Non-traditional machining methods have the ability to quickly produce complex textures in any material and they are effective against both traditional machining methods and textures created by hardening devices.

Table A2, which is presented in Appendix A, is related to this section, and in it, an effort was made to present in an organized and interesting way the application of different texturing methods, materials used, shapes of the created textures and the results obtained in each case.

Due to the ease that certain methods propose to elaborate on different forms of textures, it is necessary to know which parameters are necessary to correctly describe the topology of the features in relation to the textured surface.

The following section gives a short introduction and recommendations to correctly describe textures and their elements, as well as real examples of the most common texture shapes on textured cutting tools.

## 3. Cutting Tool Surface Texturing

The formation of controlled textures on surfaces that are subject to mutual sliding began to be used many years ago with the aim of improving the quality of lubrication in the sliding contact zone; for example, in internal combustion cylinders [59]. The term “*surface texturing*” can be explained as the deliberate formation of engineered surfaces that contain multiple features, such as microgrooves, discrete micro-dimples, discrete micro-holes, micro-protrusions, and micro/nano-patterns, as well as the combination of different shapes that forms complex patterns. There are two main functions fulfilled by textured surfaces: they can entrap wear debris resulting from rubbing, and they act as reservoirs that provide lubricating material to the contact interface when starved lubrication occurs. The joint action of these two functions allows the minimizing of third-body abrasion.

In addition to the lubrication effect, the resulting material after texturing, which is placed between the formed grooves, “*cushions*” the counter body during its movement on the textured surface. The presence of these grooves reduces the contact area, which improves the quality of the friction process. Moreover, with an appropriate design of the structured “*cushioning islands*” a reduction of wear can be reached.

Thanks to the current technological level that currently exists in the field of micro/nano-texturing, it is possible to fabricate engineered surfaces with controlled wettability, high wear resistance, low coefficient of friction, etc. This allows the creation of new applications simply by creating surfaces with specific textures that have the ability to improve the tribological characteristics (friction and wear) of a sliding process [60].

### 3.1. Texturing Parameters

Microgrooves, dimples, protrusions or their combinations are the possible configurations, which represent the building block of texturing. Designing a textured surface involves placing an element with a primitive geometric shape (ellipse, hemisphere, cone, etc.) in an array over the target surface area. Thus, to describe a surface with such structure, it is necessary to define the feature geometry, its location in the array, and the location of the array relative to the entire target surface. Thereby, parameters that can describe the texture element geometry and the topology of the features on the textured surface are needed.

Gropper et al. [61] explained in a very illustrative way what parameters are commonly used in hydrodynamic lubrication studies to describe dimples as an individual texture element. Although the authors used circular and rectangular dimples as the elements of textured surfaces (Figure 4), their explanation is applicable to any other texture shape. As Figure 4 shows, the texture element can be characterized by its size (base shape dimensions and depth), three-dimensional shape (base shape form and bottom profile), and orientation with respect to the direction of sliding in the case of an asymmetric texture.

The base shape form is the geometrical shape on the sliding surface, while the bottom profile is the texture shape inside of the material. In Figure 4a,b, the base shape dimensions of the rectangular dimple are represented by “*l*”, while this letter also symbolizes the texture’s diameter of the circular texture element. When the base shape used is not equilateral its dimensions can be represented by the length “*l*” and width “*w*”. Similarly, in these figures, the texture element depth is represented by “*h_texture_*”. The parameters “*l*” and “*h_texture_*” illustrate the maximal dimension and depth of the texture element in its sliding direction, respectively. From these parameters, the “*texture aspect ratio*” can be calculated, which is defined as “*λ*
*= h_texture_*/*l*”. Although this relationship is constantly used in many studies, its value does not capture the base shape of the texture element or its internal structure. In Figure 4, the texture element area is represented by “*A_texture_*”. Moreover, here it can be seen that each element has a peripheral area, which physically separates it from neighboring elements. Thus, the area that includes the “*A_texture_*” and the peripheral area is named the texture cell area and is represented as “*A_cell_*”. For a square cell as shown in Figure 4, its area can be calculated using the maximum cell dimensions *X_cell_* and *Y_cell_* in the following relationship “*A_cell_” = “X_cell_ · Y_cell_*”.

Moreover, for a global scale, a textured surface can be described by its “*relative textured ratio*” *B = A_textured_*/*A_c_ = α∙β* (where *A_textured_* is the textured surface area, *A_c_* is the total sliding surface area) and its “*relative texture extends*” in x- and y-direction, which is given as *α = X_textured_*/*X_c_* and *β = Y_textured_/Y_c_*, respectively (Figure 4c).

The location of the texture elements in relation to each other and their orientation with respect to the sliding direction are also of great importance in the design and description stage of a textured surface.

Another parameter, which is usually used for texture characterization, is texture density. Texture density in general can be calculated by *ρ_texture_*
*= N A_texture_/A_textured_*, where *N* is the total number of texture elements. For uniformly distributed textures, as a special case, the texture density can be calculated by the simplified formula relation *ρ_texture_ = A_texture_/A_cell_*.

One more important parameter is the “*relative texture depth*”, which can be determined by *S*
*= h_texture_/h*_0_, where *S* is the *relative texture depth*, and *h*_0_ is the minimal distance between the sliding surfaces (Figure 4c). It is necessary to take into account that sometimes instead of expressing the relative dimple depth, the “*height relation*” is used, which is represented as *H_r_*
*= (h_texture_ + h*_0_*)/h*_0_
*= S + 1*, where *H_r_* is the height relation.

Thus, the authors of the work [61] pay attention to the fact that texture aspect ratio, texture density, relative dimple depth and for partial texturing (when *A_textured_ ≠ A_c_*) the relative texture extensions are the only parameters having the most pronounced influence and are thus the most important design parameters for textured surfaces. Therefore, the studies carried out in the study of textured surfaces should contribute to finding optimal values for these parameters. In addition to the description of linearly distributed textured surfaces, this paper also gives a definition of the parameters necessary to describe partial texturing, as well as the relative texture extensions in the circumferential and radial direction, whenever polar coordinates are applied.

Abdel-Aal et al. [62] showed that in addition to “*texture aspect ratio (λ)*” and “*relative textured ratio (B)*”, there are two more parametric relations that must be considered during the design of a textured surface. They are the “*dimple slenderness ratio*” and “*surface aspect ratio*”. The first one is defined as the ratio of the height to the diameter of the texture element and the second one is defined as the ratio of the centerline-to-centerline spacing between texture elements to the height of the texture element, Figure 5.

### 3.2. Most Common Texture Shapes on Textured Cutting Tools

At the end of the 1980s in the last century, Kuznetsov et al. [63] firstly proposed to form microtextures on a cutting tool when they filed a patent, in which it was suggested to make helical and spiral grooves on the deforming and the cutting wedge’s rake and flank faces of a broach, respectively. The technical solution of this document was to increase the durability of the cutting tool with a corresponding improvement in the processing, obtaining a reduction in cutting force and, consequently, less wear on the cutting part.

Since then, a large number of investigations related to the formation of textures on the cutting tool surfaces and to the study of their influence on the machining process of different materials using diverse tool materials have been carried out. The aim of these works was to determine the optimal parameters for the texture’s design. Despite this, reports in the literature provide conflicting information. There are many reasons for the conflicts, but the main contribution is related to the following two factors: the lack of standardization of testing methods, and the absence of uniformity in reporting the recommended values. The lack of uniformity for a detailed and systematized description of the designed textures on cutting tools leads the authors of these works to omit important information about the texture parameters when writing their papers. Moreover, the almost infinite possible choices of the texture element geometries that may be chosen for texturing is another serious difficulty that leads to conflicts or to the inability to generalize the results obtained.

Despite this, most works prove that textured cutting tools have a great influence on the contact phenomena between the cutting wedge and the workpiece material. As in hydrodynamic lubrication studies, textured surfaces on cutting tools provide a reduction in the friction coefficient during cutting. Furthermore, it is possible to find diverse texture base shapes with different parameters for textured cutting tools.

In order to better define the texture surfaces used in the cutting tools, we propose a way in which to classify them according to the following parameters:Geometric dimensions scale of texture element (micro- or nanotexture);Texture element configuration relative to surface baselines (protrusion, dimple, microgrooves);Type of texture element (continuous or discrete);Base shape of continuous texture element (linear, sinusoidal, wavy, circular, squares, elliptical, complex, etc.);Base shape of discrete texture element (dimple or protrusion with square, rectangular, triangle, circle, elliptical, chevron-like, micro pyramid, hybrid shape, or other);Number of texture guide axes (uniaxial, multiaxial);Texture placement on cutting wedge (chamfer, rake or flank face);Textured surface density (full textured, partial textured);Condition of textured surfaces (coated, empty, filled with solid lubricant);Texture bottom shapes (semicircles, parabola, rectangles, squares, triangles, trapezoids, sinusoidal, curvilinear);Texture distribution type (normal square (grid) distribution array, shifted α degree distribution array);Texture orientation relative to the main cutting edge (parallel, perpendicular, oblique);Presence of gap from the cutting edge.

Table A3 (see Appendix A) shows images of several designed textured cutting tools according to a previously proposed classification.

### 3.3. Conclusions of Section 3

This section explained the term texture, as well as the parameters that correctly and completely describe the geometry of the texture elements and the parameters that describe the topology of the features relative to the textured surface. In addition, the role of textures in lubrication and wear resistance on the cutting tool surface was discussed.

Here, it was seen that texturing parameters can be summarized in several terms, such as dimensions, scale, shape, texture sizes, texture location and texture direction, which were presented as the main parameters. Knowledge of these parameters is necessary to have a clear idea of their influence on the machining process.

In order to better define the texture surfaces used in cutting tools, it was proposed to classify them according to different parameters. Moreover, Table A3 shows real examples of textures used to carry out the machining process is presented. This table shows images of the used textures, information on their geometric parameters and machining conditions for each of the cases presented. We believe that this information will be very helpful for researchers related to the area of texturing cutting tools.

The study of more than 100 articles in the area of textured cutting tools showed that different textures affect the machining process in different ways depending on cutting conditions and workpiece/tool material. The following section summarizes the results of the research on ceramic and superhard cutting tools in order to improve the machining process.

## 4. Effect of Surface Texturing of Ceramic and Superhard Cutting Tools

Ceramic and superhard cutting tools are designed for the machining and finishing of superalloys and difficult-to-cut materials. Ceramic cutting tools are desirable tools because they can withstand high temperatures even more than carbide tools [64]. Moreover, these kinds of tools possess unique mechanical properties, high hardness, corrosion resistance, low wear rate [65], high resistance to cratering and abrasive wear, and the possibility of increasing the machining cutting speed [66] in comparison with other cutting tool materials. Basically, the ceramic cutting tools are based on two types of ceramic; aluminum oxide Al_2_O_3_ (alumina) and silicon nitride Si_3_N_4_. Several materials such as silicon carbide (SiC), titanium carbide (TiC), and titanium nitride (TiN), are added into the Al_2_O_3_ or Si_3_N_4_ matrix in order to improve the mechanical properties of the ceramic cutting tools. Si_3_N_4_-based ceramics are used in cutting tools called SIALON that came into use during the 1980s and have superior characteristics to Al_2_O_3_-based ceramics when some factors are considered [67,68]. For instance, the flexural strength of Si_3_N_4_ can be in the range of 700–1100 MPa. Moreover, Si_3_N_4_ cutting tools have excellent fracture toughness and thermal shock resistance, and also exhibit stable cutting performance because they have a high crack resistance [69].

Superhard cutting tool materials are the materials, whose hardness exceeds 40 GPa on the Vickers’ hardness scale [70]. In addition, these materials show superior properties including wear resistance, high thermal conductivity (higher than 800 Wm^−1^K^−1^ [71]), high and chemical stability. Diamond and cubic boron nitride (CBN) are among the hardest materials, and their hardness is in the range of 59–75 GPa for chemical vapor deposition (CVD) diamonds, 40–80 GPa for polycrystalline diamonds (PCD), and 28–44 GPa for Polycrystalline Cubic Boron Nitrides (PCBN) [72]. In addition to diamond and cubic boron nitride, superhard materials include tetraboron carbon (B_4_C) and borocarbonitrides (B_x_C_y_N_z_); but, in the field of cutting tool materials of diamonds, polycrystalline diamond, and polycrystalline cubic boron nitrides are commonly used.

In the following subsections, a summary of the studies carried out on the influence of ceramic and superhard textured cutting tools on tool wear, coefficient of friction, adhesion, cutting force and cutting temperature will be made.

### 4.1. Effect of Texturing in Ceramic Tool on Friction Coefficient, Tool Wear and Adhesive Property

One of the first studies on the tribological properties of textured ceramic surfaces by carrying out sliding wear tests was accomplished in 2013 by Xing et al. [73]. In this study, the authors used samples of hot-pressed Si_3_N_4_/TiC ceramic and AISI440C stainless bearing steel ball as the test materials. In the ceramic samples, wavy and lined grooves were made by laser surface texturing. Moreover, the microgrooves were filled with molybdenum disulfide (MoS_2_) solid lubricants (diameter of 2 μm). The groove width, the texture spacing, and the angle of the wavy pattern were 50 μm, 200 μm, and 90°, respectively.

The results showed that the friction coefficient of the wavy grooved and lined grooved surfaces without MoS_2_ solid lubricants is lower than smooth surfaces without MoS_2_ solid lubricants, and it reaches a steady state with less time. For smooth surfaces, the friction coefficient stabilizes at a value of 0.6–0.7, while for textured surfaces, it stabilizes at 0.4–0.6. Here, the wavy grooved surface without MoS_2_ solid lubricants sample shows the lowest friction coefficient and wear rate.

After this publication, Xing et al. [74] published a study about the cutting performance and wear mechanism of nanoscale and microscale textured Al_2_O_3_/TiC ceramic tools in dry cutting of an AISI 1045 hardened steel. This was the first work carried out on an aluminum oxide-based cutting tool with a textured surface since previous studies had been carried out mainly on cemented carbides and high-speed steel tools due to their availability and ease of texturing. Here, the authors reported new ceramic tools with micro- and nanotextures on their surfaces. Furthermore, the authors investigated some self-lubricated tools, the textures of which were filled with MoS_2_ solid lubricants (diameter of 50 nm). All experiments were carried out as dry cutting tests on hardened steel. During experiments, the surface roughness, cutting temperature, friction coefficient, chip topography, cutting force, and tool wear, were measured. Moreover, the effect of different types of textures was investigated. The cutting material used for this study was a hot-pressed Al_2_O_3_/55 TiC (wt%) ceramic. In this study, three different micro-textures were generated on the rake face: wavy grooves (AT-W), linear grooves parallel to the cutting edge (AT-PA), and linear grooves perpendicular to the cutting edge (AT-PE). Furthermore, the nanotexture direction was parallel to the cutting edge and the microgrooves had a 3D topography of triangular shape. The texture width, depth and period were 40–50 μm, 45–50 μm and 150 μm. The width of the nanoscale groove was about 350–400 nm, the depth was about 120–150 nm, and the period was about 750 nm. Moreover, the nanoscale grooves were parallel to the main cutting edge. The experiments showed that the average cutting forces of three kinds of textured self-lubricated tools were reduced compared with conventional tools (AS) under the same cutting condition. The friction coefficient analysis at the tool–chip interface at different cutting speeds indicated a decreasing trend of friction coefficient in the order of AS (conventional), AT-PE (linear grooves perpendicular), AT-PA (linear grooves parallel) and AT-W (wavy grooves) tools, Figure 6. So, the maximum reduction in friction coefficient was obtained by the wavy grooves tool, and it was up to 15–20% compared with the conventional tool. The wear analysis showed that the surface damage of the conventional tool was in the form of abrasive wear, forming significant crater wear at its rake face. Moreover, a large amount of ploughs at the rake face and chamfer were observed. On the other hand, the linear grooves perpendicular to the cutting edge tool showed milder abrasive wear and crater wear at the rake face compared with the conventional tool. In addition, the tools with linear grooves parallel to the cutting edge and wavy grooves had a similar smaller crater wear at the rake face, and few ploughs can be seen on the wear area.

The EDX analysis of the rake face indicated that molybdenum disulfide was removed from micro-textures at the tool–chip interface and formed a lubricating film on the rake face. Thanks to these results, the authors deduce that the reduction in the friction coefficient is due to the presence of the formed lubricant film on the worn surface since the MoS_2_ has lower shear stress than the substrate. Therefore, it was concluded that tool–chip contact length is reduced due to the simultaneous presence of micro- and nanotextures on chamfer and rake face embedded with MoS_2_ solid lubricants. This reduction in the contact length results in a lower friction force. In addition, the authors found that the geometry of the textures had a high influence on the reduction in cutting force, friction coefficient, and cutting temperature. For instance, the wavy grooves tool was the best in improving the lubrication conditions. On the other hand, the linear grooves perpendicular tool showed the worst improvement but still showed an improvement compared to the conventional tool. The authors explained this observation by the fact that in the case of linear grooves perpendicular tools the lubricants cannot be effectively stored in the grooves, and they were easily taken away by the chips. This is due to the fact that the direction of the grooves was parallel to the chip flowing direction. At the same time, when the grooves are parallel to the cutting edge the chip flowing direction was perpendicular to the direction of grooves and the lubricants were stored in the grooves, but they were not easily released from the microscale grooves. A combination of these two facts was shown in the tool with wavy grooves, in which the largest reduction in the tool–chip contact area takes place because the lubricants can be released from the microscale grooves to the rake face.

In another work, Xing et al. [40] continued their research, but here, studying the influence of different nanotextures covered with a soft-coating WS_2_/Zr composite in dry cutting of AISI 1045 hardened steel. The cutting material was a hot-pressed Al_2_O_3_/55TiC (wt%) ceramic. Three different types of nanotextures (AN-PE (perpendicular to the main cutting edge), AN-PA (parallel to the main cutting edge), and AN-A (areal textures)) were used on their rake face. For the cutting tools with areal textures and microgrooves parallel to the main cutting edge, the nanotextures grooves were parallel to the main cutting edge. For cutting tools with microgrooves perpendicular to the main cutting edge, the grooves of nanotextures were perpendicular to the main cutting edge (Figure 7). It was noted that the width, depth and period were about 350–400 nm, 120–150 nm, and 700–800 nm, respectively. The soft-coating WS_2_/Zr composite was deposited by medium-frequency magnetron sputtering together with multi-arc ion plating. The sizes of the texture were 0.05 mm width, 0.1 mm pitch y, and 0.7 mm length, while the areal sizes were 0.7 × 0.6 mm. The analysis of the results showed that the friction coefficient of the conventional tool with WS_2_/Zr soft-coating was lower than that of the conventional tool without soft-coating, Figure 8a. Furthermore, the conventional tool without soft coatings showed the most wear, while the covered tool showed the least damage. For the last tool, the WS_2_ film was completely removed, leaving the Zr layer almost intact. Therefore, the lubricant film was not effective for the conventional tool because this coating was removed immediately during machining. In contrast, the coated nanotextured tools showed a lower coefficient of friction compared to the two conventional tools, and the covered cutting tool with areal textures showed the lowest friction coefficient. Large amounts of adhesions of workpiece materials were adhered to the tool tip of the tools with areal textures and microgrooves parallel to the main cutting edge, while their worn surfaces did not show chipping. The use of the coating on the nanotexturized tool further reduced the tool–chip contact length with the untextured tool. This led to a large reduction in cutting temperature, coefficient of friction, friction force, and shear force, Figure 8b. From the obtained results, the authors found that the geometry of the nanotextures has a great influence on the cutting performance. Thus, the nanotextures parallel to the cutting edge showed better results than the perpendicular textures. On the other hand, the nanotextured bands had less effect than the areal nanotextures. The authors explained that the nanotextures increased the adhesion strength between the coatings and substrate; while the coating was easily removed by chip flow over the tool surface with textures perpendicular to the cutting edge compared to textures parallel to the cutting edge. This is related to the fact that the nanogrooves were generated parallel to the direction of chip movement. In the case of areal textures, the nanotextures were generated in a direction perpendicular to the flow of the chips, which helped the distribution of the lubricant coating, thus achieving a greater reduction in tool–chip contact area.

Feng et al. [75] studied the influence of in situ formed textured ceramic tools for the dry cutting of an AISI 1045 hardened steel. In this work, Al_2_O_3_ and TiC were selected as the ceramic materials. In addition, the micro-textured tool was filled with a graphite lubricant. Three different types of textured tools were generated on the rake face (pit diameter variation textures (MP-D), groove spacing variation (MG-S), and groove width variation (MG-W)). Thus, the texture depth was set to 0.1 mm, and the microgroove width was 75 μm. Similarly, the distance between microgrooves was fixed at 75 μm in order to study the influence of microgroove width on the cutting performance. The prepared powder and the punch with a specific shape were cold formed in the sleeve. After cold pressing, the sleeve was sintered in a vacuum hot-press furnace. The idea of this method is based on manufacturing a surface texture during sintering, to avoid the use of micromachining methods as is usually done. In this study, the distance between the cutting edge and the textured area was 200 μm. The analysis of the results showed that tool breakage of the conventional tool occurred near the tip and a large amount of adhesion occurred on its rake face, and the existence of the plough phenomenon was observed under a high-power microscope. The rake face of the tool with groove width and groove spacing variation shows that the chip adhesion was mainly distributed in the first three textures near the main cutting edge. On the other hand, the chip adhesion on the tool with pit diameter variation occurred near the tip. The obtained results during the comparation of textured and conventional tools showed that the chip adhesion phenomenon in textured tools was reduced, and there was no obvious tool damage. Thus, the tool with groove width variation showed the least amount of chip adhesion and in this case, the graphite lubricants were stored more efficiently compared to the other two textures studied. Moreover, this cutting tool showed the best performance compared with the other textured tools. The authors concluded that tool–chip contact area was reduced owing to the texture; and, in addition, they focus on the fact that the parameters of the micro-texture considerably influenced the cutting performance. Therefore, the authors found that the tool with groove width variation had better cutting performance than the tools with groove spacing variation and pit diameter variation at different cutting speeds. The scholars explained this fact by the perpendicular direction of the grooves to the direction of chip flow in the tool with groove width variation. This allowed the lubricant to be stored and released more efficiently, greatly reducing the cutting force, compared to the other textures. Moreover, it was found that the improvement in cutting performance was not very significant in the study of the tool with groove spacing variation, where the direction of the microgrooves was perpendicular to the flow of the chips. In the case of the pit diameter variation, the stress contact region was less than in the microgroove width texture tool, however, the graphite lubricant was not effectively stored in the hole and was easily pushed out of place by the chip. As was observed in previous works, the wear morphology of the different textured tools showed that the textured tools can reduce adhesion and improve cutting performance. In conclusion, the authors observed that microgroove and -pit tools effectively reduce the tool–chip contact length, as well as usefully containing abrasive debris and storing the graphite lubricant to improve cutting performance.

Duan et al. [76] also studied the improvement of the cutting performance of Al_2_O_3_-TiC ceramic cutting tools. Here, the authors used different textured hot-pressed Al_2_O_3_/55 TiC (wt%) ceramic cutting tool, but with the implementation of an emulsion-type flood cooling during the machining of an AISI H13 steel. Two types of textured tools were used: a multiscale textured tool (MTT), and a textured tool with a microscale groove pattern (MT). Textures were especially designed to be below the rake face in order to impede the derivative cutting in the microgrooves, Figure 9. At the same time, the location of the micro-textures on the rake face is the same for the MT and MTT tools.

For the tool with a microscale groove pattern, significant microcracks in the cutting nose zone and flaking of the tool material were observed. Moreover, the SEM analysis showed that its microstructures were blocked, and they were filled with the workpiece material. Even so, small traces of the used emulsion-type cutting fluids were observed in the textures, which demonstrated that only a small amount of cutting fluid permeated the cutting zone. In the case of multiscale textured tools, no chip material located in the textures was observed, on the contrary, the features were easily recognizable. Furthermore, in the cutting nose zone, no obvious defects were found. Thus, this tool showed significantly less adhesive wear, and the amount of cutting fluid that permeated the cutting zone was larger, compared with the MT tool. Additionally, the MTT tool had better lubrication and wear resistance than MT, and for that, no crater trace was found on the multiscale textures. The adhesions on the rake face of the MTT were less than those on the MT. Finally, the scholars concluded that multiscale textured tools exhibited better wear resistance at different cutting velocities. Furthermore, this type of texture promotes permeation and the introduction of cutting fluid at the tool–chip interface, and at the same time impedes the derivative cutting, which effectively enhanced the cutting performance of the Al_2_O_3_/TiC ceramic tool. On the contrary, the derivative cutting in conventional microgroove textures led to the intense formation of texture locks, making the use of these tools ineffective.

In the work [77], it was attempted to improve the tribological behavior of a tool during turning AISI 52100 hardened steel by using micropatterned inserts. The insert was a superhard CBN cutting tool, and the pattern was formed on the rake surface using layer-by-layer electrical discharge machining. To obtain lines with a width of ∼110 μm and a depth of ∼50 μm microelectrode with a diameter of 90 μm was used. The results calculated by measuring wear and modeling continuous and saw-chip formation at different feed rates and surface velocities showed that friction of the micropatterned insert was reduced by 9.5∼34.5% at different feed rates compared to regular tools (Figure 10a). Meanwhile, for various surface velocities, friction was reduced by 6.4∼34.3% due to micropatterned inserts (Figure 10b). At low feed rates and surface velocity, the beneficial influence of micropatterning is observed the most. It was also reported that flank tool wear was reduced by 9.7∼11.4% for the proposed tools and the wear crater differed a little. Furthermore, the authors indicate that the cause of the reduced friction is the presence of air gaps in the micropatterned insert during machining. It was assumed that they increase shear stress and also improve debris elimination from the contact area uniforming the contact pressure. Finally, the authors concluded that the friction is reduced because the micropatterns on the CBN inserts distribute the contact stress of tool chips effectively, they produce more shear than non-patterned tools, and prevent abrasive wear and the debris in the cavity area.

In another work, Li et al. [78] investigated the cutting performance of micro-textured superhard polycrystalline cubic boron nitride (PCBN) cutting tools during the machining of GCr15 hardened steel. In the inserts, two micro-hole textures with different diameters (80 µm and 120 µm) and with a depth of 5 μm were formed on the tool rake face. First, the study performed finite element analysis (FEA) to predict the role of the micro-hole textures play in alleviating tool wear. It was found that inserts processed with the d = 80 µm tool have the lowest contact area at the tool–chip interface (Figure 11). Thus, the wear observed in the micro-hole tool with d = 80 µm was less than that of the micro-hole tool with d = 120 µm, Figure 12. Furthermore, by measuring shear and Mises stresses the influence of surface texturing on tool wear was demonstrated. For instance, the maximum shear and Mises stresses for the micro-hole tool with d = 80 µm were 1337 MPa and 2376 MPa, respectively; while these values for the micro-hole tool with d = 120 µm were 1443 MPa and 2517 MPa, respectively, Figure 13. On the contrary, the maximum shear and Mises stresses for the non-textured tool were 1512 MPa and 2620 MPa. Therefore, these results indicated that the fatigue behavior of the tool could be evaluated by the stress state of the tool. Thus, the lower stress indicates that the degree of tool surface wear should be lower. At the same time, the authors were able to find correlations between tool wear, tool surface stress and the cutting force. It was concluded that low cutting force and tool surface stress promotes a longer life cycle of the tool. The cutting experiments carried out in this work confirmed the predictions obtained from the finite element analysis.

An interesting work was published by Wang et al. [79], in which the cutting characteristics and friction of micro-textured single-crystal diamond tool were carried out on an oxygen-free copper disk. With this study, the following five different types of textures were formed on the single-crystal diamond block by femtosecond laser: linear grooves parallel to the cutting edge (PGT); linear grooves vertical to the cutting edge (VGT); concentric circular texture (CT); ring sequence texture (RT); mesh texture (MT). The patterns were made 10 μm away from the cutting edge covering the area of 80 × 60 μm^2^ with a 100 nm depth. The friction coefficients of the micro-textured diamond surface were calculated from the friction forces, which were previously obtained by a test system. The results of these tests show that the friction coefficients of the micro-textured surface under dry friction compared with the single-crystal diamond block without micro-texture were increased except for the surface with concentric texture. Adding lubricant drastically improved the tribological behavior of the specimens. Thus, surface texturing was not able to outperform regular tools under dry friction conditions. The friction coefficients of the micro-textured tools showed a decreasing trend under the lubrication condition during cutting experiments, which is in concordance with the trend observed in the test system. Therefore, the results of the tests prove that texturing the surface can reduce the friction coefficients of the rake face in lubrication conditions. At the same time, the rake face and wear depth of the regular tool were measured to be relatively large, which could be explained by the presence of high pressure at the tool–chip interface, which does not allow the cutting fluid to penetrate and perform its lubrication function, resulting in poor friction reduction effect. On the contrary, the PGT tool shows the lowest wear area and wear depth. It can be observed that other textured tools also had lower wear area and depth at the tool–chip interface since surface texturing improves the friction condition between the tool and chip due to the formation of a lubrication film on the tool rake face. For instance, for the PGT tools, lubricant normally moves away from the cutting edge. However, due to the destruction of the groove edge, fluid may be partially redirected towards the cutting edge, filling the plane between the grooves and forming a stable lubrication film. In the case of VGT, the lubricating fluid is expected to flow away from the cutting edge affected by chips. In fact, subsequent chips make the fluid flow further to the far side of the cutting edge. For CT and RT tools, it is more likely for the lubricant to move through the texture groove towards the middle areas being further collected on the bottom of the ring. Compared to CT tools, the RT type is anticipated to have a more dispersed lubricating film. Mesh textured ones will be divided into several independent blocks which will be filled with lubricant preventing total spreading on the surface. Finally, the authors concluded that stable lubricant film and, hence, improved tribological properties are possible only for the PGT type while in other cases this improvement is achieved due to filling the tool–chip interface. Thus, such tribological parameters as the cutting forces, average friction coefficient, and tool wear were reduced more which resulted in an effective improvement of the friction performance of micro-textured tools.

Another study, conducted by Ghosh and Pacella [80] focused on the performance of textured polycrystalline diamond (PCD) cutting tools during dry machining of an aluminium alloy 6082. Within the work, three types of surface textured tools were used: grooves perpendicular to the chip flow direction (CFD), grooves parallel to CFD (at an acute angle inclined to the direction of the chip flow) and grooves parallel to the main cutting edge (MCE). Moreover, a regular non-textured cutting tool was used as a reference to compare the results (Figure 14). Geometrical parameters of the grooves were: 260 nm in depth, 7 μm in width and spaced at a pitch (center to center) of 20 μm. The friction coefficient at the tool–chip interface was calculated using the average cutting force and it is shown in Figure 15. The friction coefficient of the reference tool was found to be in the range of 0.08 to 0.6 while for parallel to CFD it was measured to be between 0.14 and 0.42.

The friction coefficient for textured grooves perpendicular to CFD showed values between 0.14 to 0.65. Specimens parallel to MCE demonstrated the poorest tribological behavior. As a result, it was stated that the friction coefficient values of the tested tools were higher compared to the reference tool.

Based on the results obtained, the authors conclude that texture shape and the groove direction influence the chip formation mechanism and chip flow. For instance, the best cutting performance was observed in the tool with grooves perpendicular to the flow direction, while textures parallel to the chip flow direction reduced the friction in the tool–chip interface by 14.28%.

### 4.2. Effect of Texturing in Ceramic Tool on Cutting Force

The cutting force analysis accomplished in the work of Xing et al. [74] showed that compared to conventional tools, the average cutting forces of textured tools were reduced, Figure 16. In this figure, it can be seen that the fluctuation of the cutting force of the textured tools was less than the fluctuation observed in the conventional tools. In addition, here it is appreciated that the cutting forces of textured tools decreased with the increasing cutting speed. For instance, its axial (F_x_), radial (F_y_), and main (F_z_) were reduced by 20–25%, 30–35%, and 15–20%, respectively. This figure shows that the AT-W tool showed the smallest cutting force.

Feng et al. [75] also performed an analysis of the cutting forces. Here, the authors noted that micro-texture parameters considerably influenced the cutting performance, Figure 17. For instance, the cutting force of the textured tools was higher than that of the conventional tools, when the spacing of their microgrooves exceeded a certain value. The authors explained that cutting forces and friction increase with increasing microgroove spacing due to a longer contact length between the chip and the tool.

For the tool with groove width variation, the width had a considerable influence on the cutting force. For a better effect of squeezing out carbon-based lubricant and to transform the contact conditions of the rake face and chips with the chip accumulation, it is rational to use microgrooves with a large width. On the other hand, the cutting quality may deteriorate with wider grooves due to the effect of secondary cutting by the edges of the grooves, which in turn leads to an increase in cutting force. The reduction in the contact zone between the chips and the cutting tool due to the large diameter of the micro-pits and the displacement of solid grease from them leads to a decrease in friction and cutting forces. The variations in three types of cutting forces for various cutting instruments at several cutting speeds (from 60 to 240 m/min) are shown in Figure 18. According to the presented results, the cutting force decreases with increasing cutting speed.

Moreover, it was proved that the three micro-textured in situ formed tools had less cutting force than the ordinary tool. The tool with the change of groove width had the lowest cutting force compared to the conventional tool and its main cutting force was reduced by 8.3–17%, radial cutting force by 16.3–24.1% and axial cutting force by 12.6–23.7%.

Ghosh et al. [80] investigated the emerging cutting forces during the machining of a 6082 aluminium alloy with a PCD cutting tool. The details of the experiments were explained in Section 4.1. Here, fluctuations from 18 to 58 N were noted for radial forces of tools with grooves parallel to the direction of chip flow that were smaller compared to the reference tools. Moreover, due to the textured tools, a reduction in feed and traction forces at different cutting lengths was achieved (Figure 19). Compared to the reference, the average cutting forces and cutting forces in the tool with grooves parallel to CFD were lower and more stable, respectively. In addition, when the sliding distance was increased from 1 to 2.758 km, there was a stable trend in feed force (varying between 20 and 30 N) for the textured tools. Similar to the above, a uniform trend (varied between 60 and 75 N) was also observed for the thrust force for the tool with grooves parallel and perpendicular to the CFD compared to the reference (73 to 90 N) one. These findings demonstrate that low variations in feed and radial forces can be achieved with different texture designs that strongly influence the measured forces when micro-textured PCD tools are operated under dry cutting conditions. In addition, it was reported that the direction of the grooves had an insignificant impact on cutting forces compared to the reference and that the structured tools enhanced the processability of the cutting forces.

On the one hand, it was proved that when the texture was perpendicular to the direction of the texture, the thrust force decreased. On the other hand, when the direction of the chip flow, the main cutting edge and the texture coincided, the thrust force was the same or slightly less than that of the reference one. Finally, it should be noted that the cutting forces of textured tools with grooves perpendicular and parallel to the direction of chip flow were reduced by 23% and 11.76%, respectively.

According to studies conducted by Li et al. [78], it was shown that non-textured tools created greater cutting forces than tools with micro-holes. Figure 20 shows that the largest main, radial and axial cutting forces relate to a non-textured tool in comparison with the forces created by a tool with a micro-hole d = 80 µm. Thus, according to Section 4.1., the decrease in tool wear was associated with less impact of forces on it due to the existence of micro-holes that reduced the friction force between the cutting surface of the tools and the chips. Textures with micro-holes d = 80 µm were more propitious for decreasing the contact area between the rake face and the chips and reducing the friction force due to different diameters and the number of textures with micro-holes. Therefore, a close link between cutting force, tool surface stress, tool wear and tool life has been obtained.

In another paper, the authors proposed an optimized laser production process for the machining of aluminum alloy 6082 using a polycrystalline diamond cutting tool for the manufacture of micro-nano grooves at its tip [81]. The authors compared the performance of a conventional cutting tool with a tool having textures on the rake face, namely grooves perpendicular and parallel to the chip flow direction (CFD) and parallel to the main cutting edge (MCE). The depth, width and pitch of the arrangement (from center to center) of the grooves were 261 nm, 7 microns and 50 microns, respectively. It was noted that higher forces (periodic changes) showed a tool with textures parallel to CFD processing cases, while grooves perpendicular to CFD and parallel to MCE worked well in contrast to conventional cutting, that is, a decrease in the average thrust force (Figure 21). Consequently, in cutting conditions without the use of coolant, the radial force and thrust force can be provided by using micro-textured PCD tools when compared with the reference one. All textured cutting tools revealed a reduction in average cutting forces compared to the benchmark; however, insert with grooves parallel to chip flow direction demonstrated a large variation in forces as shown by the lower and upper quartiles in Figure 21.

Another conclusion made by the authors is that by using tools with grooves parallel to the direction of chip flow, it is possible to achieve synergetic effects in decreasing cutting force and improving the anti-adhesive impact.

Fan et al. [82] in their work studied the cutting performance of the PCBN tools during the turning of a heat-treated Cr12MoV alloy. Figure 22 shows the types of microstructures that were made using laser texturing on the tool rake faces. The pit diameter and pitch of the circular-pit texture were both 70 μm. The groove width, length and groove spacing of the elliptical-groove texture were 70, 150, and 70 μm, respectively. The groove width of the transverse-groove texture and the groove pitch were 40 and 70 μm, respectively. The groove width was 40 μm and the groove spacing of the composite-groove texture was 70 μm. The groove width of the wavy grooves and the groove pitch were 40 and 70 μm, respectively.

The authors carried out an FEA in order to predict the total cutting force on the tool rake faces at different cutting speeds, Figure 23a. The results indicated that when the cutting speed increases, the total cutting force on the tool decreases. Furthermore, at the same cutting speed, the total cutting force for each micro-textured tool was less than that of the conventional tool. Compared with the conventional tool, the cutting forces with circular pits, elliptic grooves, transverse grooves, composite grooves, and wavy grooves of the textured tools were decreased by 6.1–12.2%, 7.1–14.4%, 3.5–9.9%, 2.1–7.2%, and 9.7–16.1%, respectively. On the other hand, Figure 23b shows that the presence of a micro-texture has no obvious influence on the axial force F(X) but has a significant influence on the radial F(Y) and main F(Z) cutting forces. Furthermore, it was noted that the real chip–tool contact area is smaller due to the presence of the micro-texture, which explains the reduction in cutting forces.

Wang et al. [79] analyzed the cutting forces that emerged on the rake faces of the five different types of microtextures. Here, it was observed that the cutting forces of all micro-textured tools were reduced by 15–25%. For instance, the feeding forces of the linear grooves parallel to the cutting edge and concentric circular texture tools were reduced by 3.8% and 5.7%, respectively, while this force in the linear grooves vertical to the cutting edge, ring sequence texture and mesh texture tools were increased by 13.2%, 7.5%, and 18.9%, respectively. On the other hand, the main cutting forces of the micro-textured diamond tools (except for the concentric circular texture tool) were reduced compared to the conventional tool.

In the paper of Wang et al. [83], the influence of texture shape and arrangement of PCD cutting inserts on nanofluid MQL turning of an aluminum alloy 6061-T6 was studied. In this work, five types of microstructure arrangement were formed in the rake face of the cutting tools: microgrooves inclined 45° in relation to the main cutting edge (T2), cross microgrooves (T3), microgrooves perpendicular to the main cutting edge (T4), concave plane (T5), microgrooves parallel to the main cutting edge (T6), and for comparison, a conventional tool (T1) was used. The cutting feed force under seven experimental conditions. Compared with dry cutting (experiment number 1), the presence of nanofluid atomized droplets (experiment number 2) can reduce the feed force by 29.36%. The tools T2 (experiment number 3), T4 (experiment number 5), and T5 (experiment number 6), which were under the condition of nano-enhanced bio-lubricant MQL, can effectively reduce the feed force by 8.57%, 13.46%, and 8.61%, respectively, compared with the conventional tool in experiment number 1. On the other hand, the cutting forces obtained in tools T3 (experiment number 4) and T6 (experiment number 7), which were also under the MQL, increased by 18.82% and 20.93%, respectively. The radial force was measured during the cutting experiments. Here, it can be seen that the use of nanofluid MQL can reduce by 45.22% the radial force compared to dry cutting. Thus, tools T2 (experiment number 3), T4 (experiment number 5), and T5 (experiment number 6) could effectively reduce the radial force by 11.38%, 16.23%, and 16.55%, respectively, compared with the conventional tool in experiment number 1. The minimum radial force (25.51 N) was observed for the tool T5 (experiment number 6), while the force of T3 (experiment number 4) and T6 (experiment number 7) was increased by 11.55% and 10.89%, respectively. Compared with dry cutting (experiment number 1), the nano-enhanced bio-lubricant MQL reduces the main cutting force by 5.5%. As was shown later, the tools used in experiments 3, 4 and 5 could effectively reduce the main cutting force by 1.6%, 6.34%, and 2.41%, respectively. Tool T4 showed the smallest main cutting force (111.96 N). In the same way, the main cutting force of T3 (experiment number 4) and T6 (experiment number 7) increased by 2.59% and 4.23%, respectively. Finally, the authors concluded that the arrangement direction of the texture on the tool surface has a great influence on the cutting force. Thus, the textured tool with grooves perpendicular to the main cutting edge (T4) under the condition of nano-enhanced bio-lubricant MQL could obtain the lowest main cutting force (111.96 N); and showed the highest reduction in feed, radial, and tangential forces (13.46%, 16.23%, and 6.34%, respectively).

Duan et al. [76] studied the cutting force on two types of textured tools: a multiscale textured tool (MTT), and a micro-textured tool with a microscale groove pattern (MT). Figure 24 shows the average cutting forces obtained with the tools with a microscale groove pattern and multiscale textured tool over the range of cutting velocities. In Figure 24, it is seen that with increasing cutting speed to 167 m/min the cutting forces gradually decreased, due to the thermal softening experienced in the chip material; but as the cutting speed increases to 249 m/min, the cutting forces increase sharply.

At this high velocity, the increase of cutting forces might be related to the tool wear, which occurs due to the high cutting temperature caused by the velocity of 249 m/min. Furthermore, it was also observed that the three cutting force components of the multiscale textured tools were lower than those of the tool with a microscale groove pattern. Thus, compared with the microscale groove pattern tool the cutting forces of the multiscale textured tool were lower by 9.3–11.2% when the velocity was 80 m/min. On the contrary, when the velocity was 249 m/min the cutting forces of the MTTs were 10.7–14.5% less than those of the MT tools.

### 4.3. Effect of Texturing in Ceramic Tool on Cutting Temperature

Although quite a bit of research has been done on texturing in ceramic cutting tools, very few studies have investigated the effect of textures on cutting temperature. This subsection is dedicated to showing the results of three experimental studies that were carried out in this field.

Thus, Xing et al. [74] in their work demonstrated that the highest cutting temperature at a cutting speed of 240 m/min in dry cutting conditions for the conventional tool (AS), tools with wavy grooves (AT-W), linear grooves perpendicular to the cutting tool (AT-PE), and linear grooves parallel to the cutting tool (AT-PA) were 495 °C, 420 °C, 430 °C, 450 °C, respectively. The highest cutting temperature distribution at the tool–chip interface of the tool with wavy grooves was about 446 °C in dry cutting conditions at a cutting speed of 80 m/min. Figure 25, shows the tool–chip interface temperature of each used textured cutting tool by different cutting speeds. This figure demonstrated that with increasing cutting speed, cutting temperature also increased for all four types of tools, and the cutting temperature of textured self-lubricated tools (linear grooves perpendicular to the cutting tool, linear grooves parallel to the cutting tool, and wavy grooves) was lower than that of the conventional tool.

From Figure 25, it can be seen that tools with wavy grooves (AT-W) showed the lowest cutting temperature at different cutting speeds among the other tools; and this instrument reduced the cutting temperature by 10–20% compared with the conventional tool. In addition, the authors explained that the reduction in temperature may be related to the following two facts. First, the reduction in cutting heat is related to the reduction in the coefficient of friction due to the lubricated film formed at the chip–tool interface, and second, the heat radiation area and heat convection with air increased thanks to the presence of micro– and nanotexture.

In the work of Feng et al. [75], it was found that tools with groove width variation reached a temperature of 190.7 °C at the chip contact interface under dry cutting conditions (v = 120 m/min, *a_p_* = 0.2 mm, *f* = 0.102 mm/r). Furthermore, in this investigation, it was observed that the cutting temperature in all tools increased with the increase in cutting speed and that the temperature of the textured tools was always lower than that of the conventional tool, Figure 26.

Among the textured tools, the lowest cutting temperature at different speeds was observed in the tool with groove width variation. Furthermore, at higher speeds the cooling effect of this tool was apparent, and at the same time, the temperature was reduced by 29.2% compared with the conventional tool.

Finally, Duan et al. [76] in their study investigated the effect of textures on the temperature in the cutting area. In this work, the cutting temperature of the multiscale textured tool (MTT) was approximately 523.8 °C, which is lower than that of the tool with a microscale groove pattern (MT) (637.8 °C) at a velocity of 249 m/min. Figure 27 shows the increase in cutting temperatures with the increase in cutting velocity in the MT and MTT tools.

Here, it can be seen that MTT temperatures were lower than MT temperatures. For instance, the temperature difference at different speeds was in the range of 12.3–17.9% and showed an increasing trend with increasing speed. The results show that multiscale textures can reduce the cutting temperature more effectively, especially at high cutting velocities.

### 4.4. Effect of Texturing in Ceramic Tool on Machined Workpiece Roughness

Feng et al. [75] in their work investigated the machined workpiece roughness after machining with a textured ceramic (Al_2_O_3_-TiC) cutting tool. In this work, the authors determined that the workpiece roughness was better when the textured cutting tool was used. At the same time, the authors determined that the roughness value decreases with an increase in cutting speed from 60 to 240 m/min with constant cutting depth and feed (*a_p_* = 0.2 mm, *f* = 0.102 mm/r). In these conditions, the tool with groove width variation showed the lowest roughness values compared to the other textured tools.

Ghosh et al. [80] also investigated the surface roughness of an aluminium alloy (6082) workpiece after machining with a polycrystalline diamond cutting tool. Figure 28 shows the obtained workpiece roughness (Ra, Rz and Rt) after machining with textured and conventional (benchmark) tools. The surface roughness was lower (Ra, Rz and Rt, respectively, 1.2963 μm, 6.1055 μm and 6.3592 μm) when they were machined with the cutting tool having grooves perpendicular to the chip flow direction. On the other hand, the surface roughness showed an increase (Ra = 1.845 μm, Rz = 10.534 μm, Rt = 11.12 μm) when they were machined with the tool with grooves parallel to the chip flow direction.

Based on these results, the authors deduced that the decrease in roughness is due to the formation of chips with different morphology; on the other hand, the grooves parallel to the chip flow direction destroy the cutting edge’s integrity which is responsible for the worsening of the machined quality of the surface.

Fan et al. [82] investigated the influence of different micro-textured polycrystalline cubic boron nitride tools on the machined surface roughness workpiece. Figure 29 shows that the conventional tool forms the largest surface roughness Ra of the machined surface compared with the textured tools. So, the elliptical-groove micro-textured tool showed the smallest surface roughness (56.7% lower than the conventional tool). The reduction in roughness from the wavy-groove micro-textured, the transverse-groove, the circular-pit and the composite-groove tools were 54.6%, 36.1%, 29.8%, and 11.3%, respectively.

Based on these results, the authors conclude that micro-texture changes the direction of chip flow to a certain extent, improves the bonding phenomenon on the machined surface, and reduces the roughness of the machined surface.

Li et al. [78] studied the surface roughness of the workpiece machined by different cutting tools. Figure 30a shows that a micro-hole tool with a diameter of 80 µm produces a surface with the smallest roughness. In contrast, when a micro-hole tool with a diameter of 120 µm was used, the surface roughness was 1.64 µm, and its surface quality was inferior compared to the micro-hole tool with 80 µm. The non-textured tool showed the worst surface quality (1.76 µm).

The authors deduced that cutting force has a high influence on the machined surface roughness when textured tools are used. Thus, the lower the cutting force, the smaller the extrusion and shearing effect between the rake face of the tool and the surface material of the workpiece.

### 4.5. Conclusions of Section 4

Ceramic and superhard cutting tools introduce significant performance and high ability of machining difficult-to-cut materials, thus the concern of enhancing their performance presents a trend and challenge. In this section, texturing on the surface of ceramic and superhard cutting tools, as well as brief definitions and significant properties of ceramic and superhard cutting tools were reviewed and explained.

Furthermore, a literature review of various studies showed the effect of texturing on the performance of mentioned cutting tools materials and that survey is combined with significant results of each study.

In addition, this section shows that surface texturing can positively affect the coefficient of friction, tool wear, adhesive properties, cutting forces, temperature, and roughness, depending on the correct choice of cutting parameters for each material under study. Moreover, it was noted that the current challenges that mainly take place during the implementation of textures are: determining the optimal direction of “groove” textures, the decrease in strength of cutting material due to the presence of textures near the cutting edge, and the “texture blockage” due to adhesion of workpiece material in texture cavities. Table 2 shows these challenges and their solutions.

## 5. Conclusions

The current paper reviewed for the first time the development to date of textured ceramic and superhard cutting tools, highlighting their role in improving tribological properties at the tool–chip interface by the creation of micro- and nanotextures on the different tool faces. This review paper shows that the studies carried out on ceramic and superhard textured cutting tools are very few, compared to the same works on carbide tools. The ceramic tools studied to date are Si_3_N_4_/TiC and Al_2_O_3_/TiC, to which solid lubricants such as MoS_2_, WS_2_ or soft-coating WS_2_/Zr are usually applied; while polycrystalline cubic boron nitrides and diamonds are the examples of investigated superhard materials.

In Section 2 of the present work, the different techniques involved in creating textures on ceramics and superhard materials were mentioned. Section 3 describes the parameters that are generally used in surface texturing studies and that should be reported in all future studies of textured cutting tools for a better understanding of their effects on the cutting process. In order to better define the texture surfaces used in the cutting tools, Section 3 is proposed a way to classify them according to certain parameters. Section 4 highlights the effect of ceramic and superhard textured cutting tools in improving cutting performance, such as coefficient of friction, tool wear, cutting forces, cutting temperature, and machined workpiece roughness. Furthermore, this review shows that the effect of micro- and nanotextures on cutting tools dramatically improves the machining performance of difficult-to-cut materials.

The advantages of texturing ceramic and superhard cutting tools to improve the machinability of difficult-to-cut materials are as follows:-The use of textured cutting tools helps reduce cutting forces due to the reduction in the contact area in the tool–chip interface in both wet and dry cutting.-Dry machining with the application of hard lubricants on textured tools shows a greater reduction in cutting forces, compared to machining without lubricant, which improves the cutting performance. This is because the textures store the lubricant and deliver it effectively and proportionately at the tool–chip interface.-The reduction in the contact area through the use of textured cutting tools leads to a decrease in the coefficient of friction, which is improved with the application of solid lubricants.-The texture shape, its parameters (such as width, depth, edge distance and density) and its orientation have a significant effect on improving the cutting performance of the cutting tool.-The use of textured cutting tools helps reduce the cutting temperature as textures improve heat dissipation by increasing the heat radiating area.-The use of textured cutting tools showed the ability to reduce variability in cutting forces.-The reduction in adhesion in the chip–tool interfaces together with the joint manifestation of previously listed advantages leads to a reduction in tool wear compared to the non-textured tool.-The nanotextures showed a lesser adherability of the work material with the textured tool compared to the micro-textures.

This work may be useful to researchers, professionals, and fabricators in the area of ceramics and superhard cutting tools looking to improve the machining performance of difficult-to-cut materials. 

## Figures and Tables

**Figure 1 materials-15-06945-f001:**
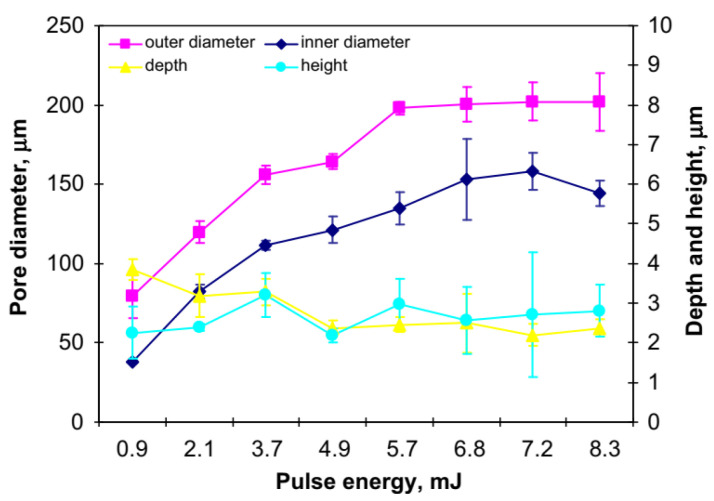
The effect of the pulse energy on the dimple diameter, depth and height. Reproduced with permission from [37].

**Figure 2 materials-15-06945-f002:**
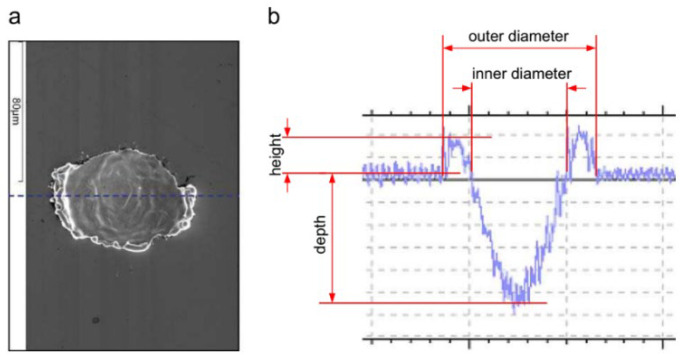
The dimple shape: (**a**) SEM micrograph and; (**b**) 2D profile illustrating the outer and inner diameter, depth and height of micro-pores. Reproduced with permission from [37].

**Figure 3 materials-15-06945-f003:**
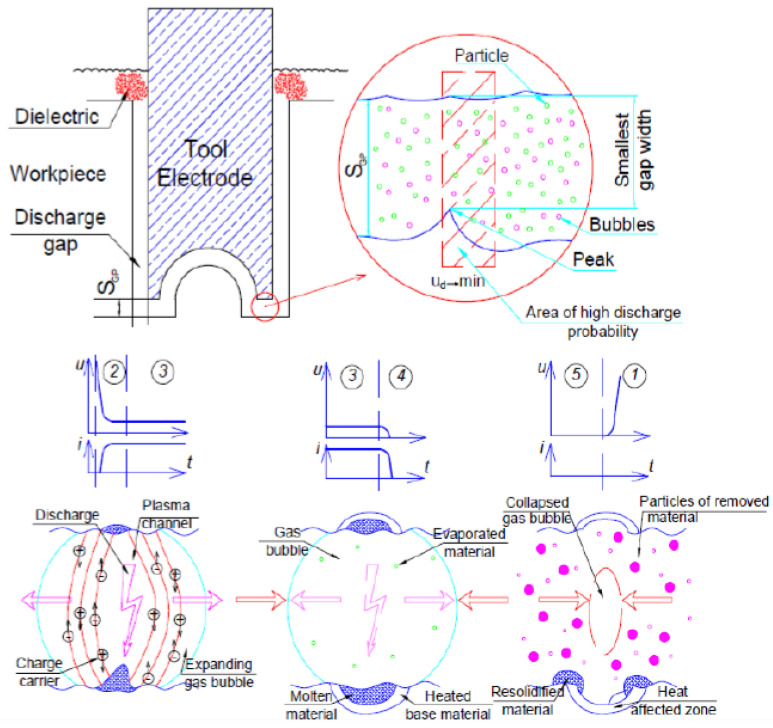
The pulse occurring through the electrical discharge machining [50].

**Figure 4 materials-15-06945-f004:**
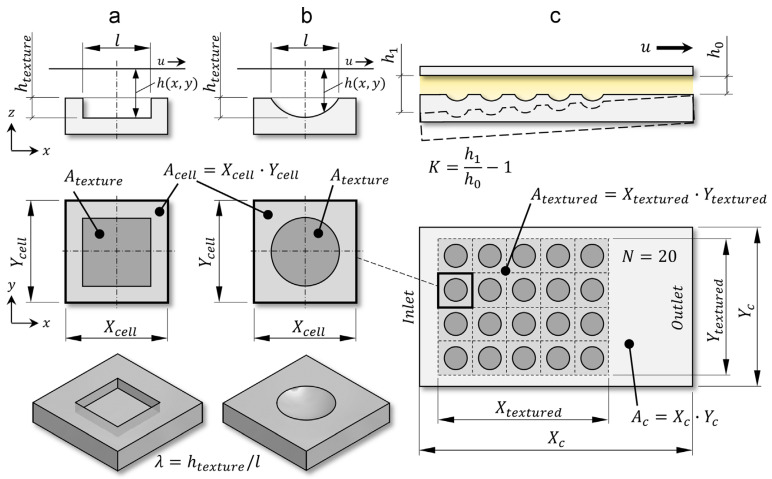
Texturing parameters for parallel/convergent slider bearings: (**a**) cuboid dimple with texture cell; (**b**) spherical dimple with texture cell and; (**c**) parallel/convergent slider bearing [61].

**Figure 5 materials-15-06945-f005:**
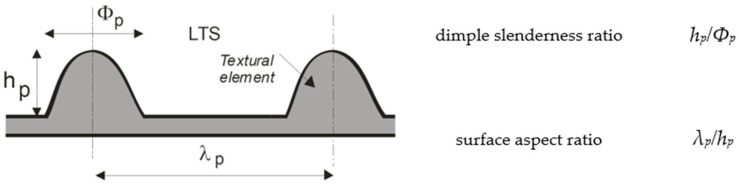
Definition of the primitive geometrical attributes for a textured surface, where *h_p_*—is the height of the texture element, *Φ_p_*—is the texture element base diameter and *λ_p_*—is the centerline-to-centerline spacing between texture elements [62].

**Figure 6 materials-15-06945-f006:**
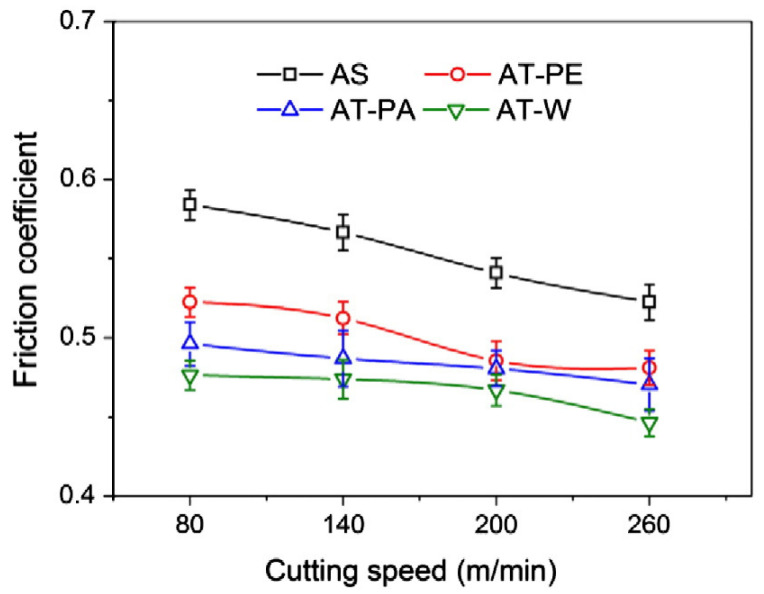
Friction coefficient at the tool–chip interface of four kinds of tools at different cutting speeds (*a_p_* = 0.2 mm, *f* = 0.2 mm/r) [74].

**Figure 7 materials-15-06945-f007:**
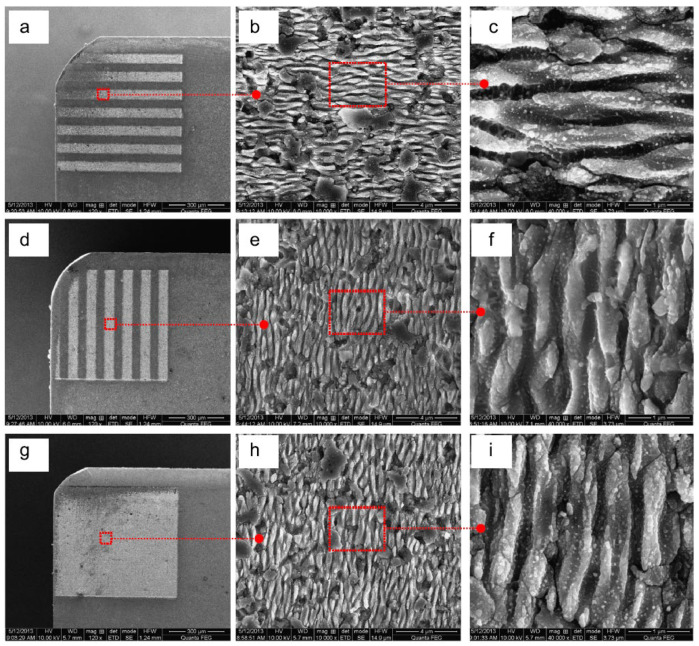
SEM images of the three types of nanotextures on the tool rake face: (**a**–**c**) AN-PE; (**d**–**f**) AN-PA; (**g**–**i**) AN-A [40].

**Figure 8 materials-15-06945-f008:**
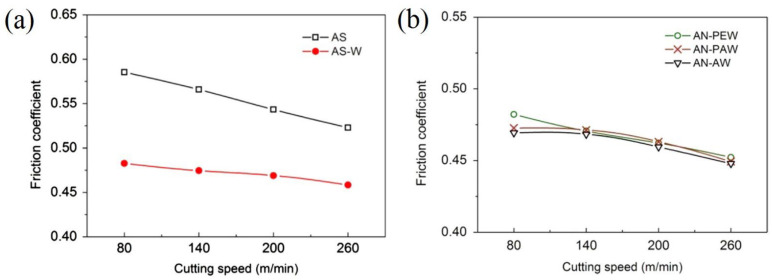
Friction coefficient at the tool–chip interface at different cutting speeds of: (**a**) conventional tools with (AS-W) and without (AS) WS_2_/Zr composite soft-coatings; (**b**) three kinds of nano-textured tools deposited with WS_2_/Zr composite soft-coatings [40].

**Figure 9 materials-15-06945-f009:**
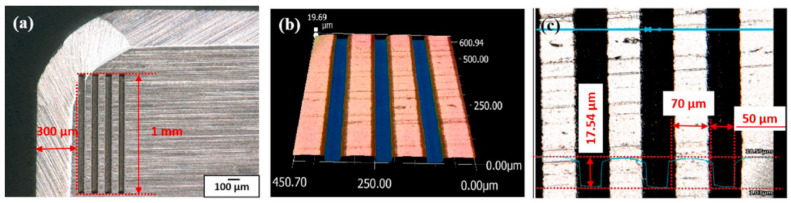
Optical and three-dimensional images of the micro-textured tool (**a**) shows the location of the microgroves on the rake face; (**b**) shows the 3D surface scanning of the microgroves; (**c**) shows the three-dimensional size of the microgrooves [76].

**Figure 10 materials-15-06945-f010:**
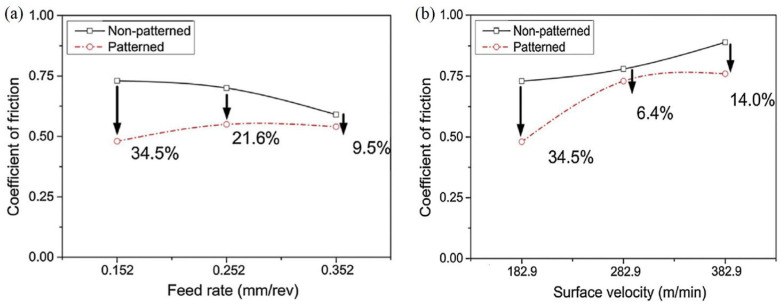
Coefficient of friction of patterned and non-patterned CBN cutting tool according to (**a**) various feed rates; and (**b**) surface velocity [77].

**Figure 11 materials-15-06945-f011:**
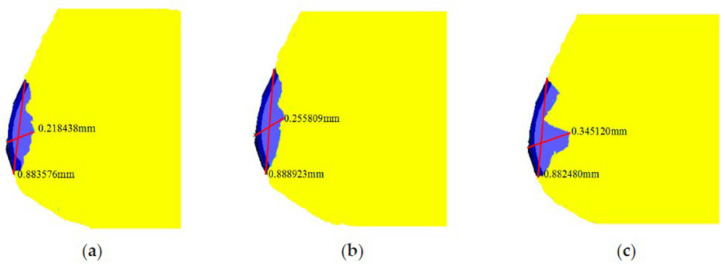
Predicted tool wear: (**a**) d = 80 μm micro-hole tool; (**b**) d = 120 μm micro-hole tool; (**c**) non-textured tool [78].

**Figure 12 materials-15-06945-f012:**
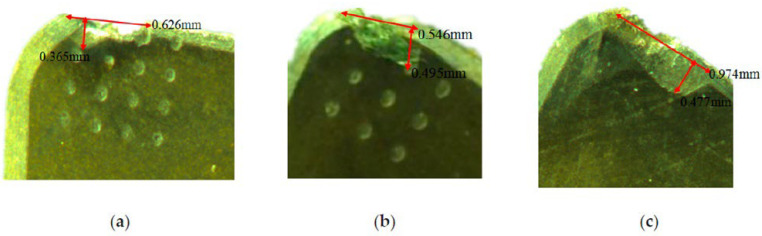
The micrograph of tool wear: (**a**) d = 80 μm micro-hole tool; (**b**) d = 120 μm micro-hole tool; (**c**) non-textured tool [78].

**Figure 13 materials-15-06945-f013:**
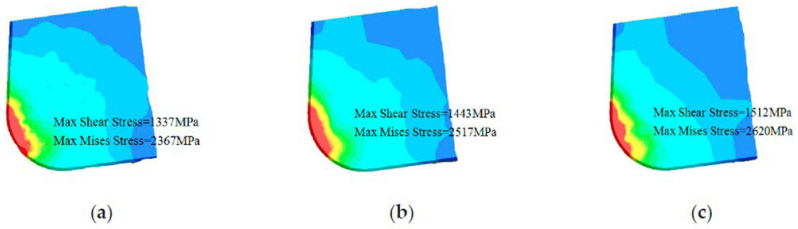
Comparison of max shear stress and Mises stress: (**a**) d = 80 μm micro-hole tool; (**b**) d = 120 μm micro-hole tool; (**c**) non-textured tool [78].

**Figure 14 materials-15-06945-f014:**
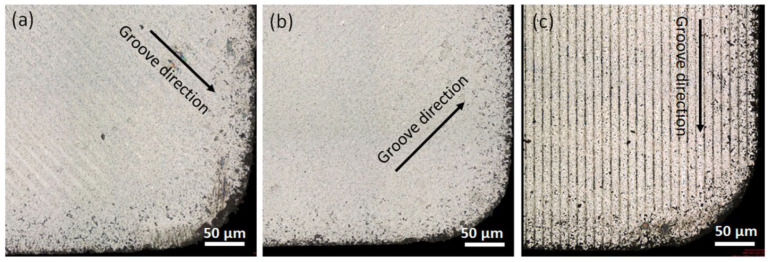
Optical microscopy of micro-nano laser processed samples: (**a**) of grooves parallel to CFD; (**b**) grooves perpendicular to CFD; and (**c**) grooves parallel to MCE [80].

**Figure 15 materials-15-06945-f015:**
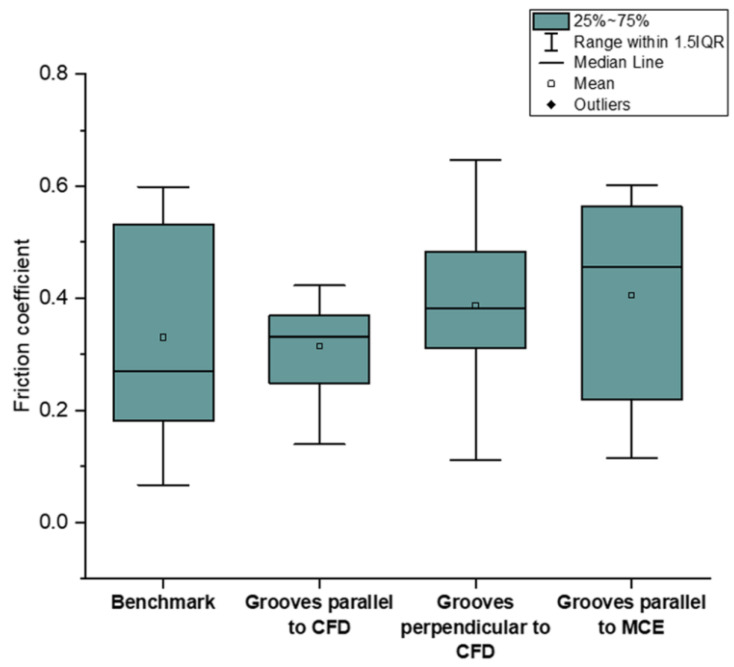
Average value of the coefficient of friction calculated from the cutting forces data for all types of machining cases benchmark, grooves parallel to CFD, grooves perpendicular to CFD and grooves parallel to MCE [80].

**Figure 16 materials-15-06945-f016:**
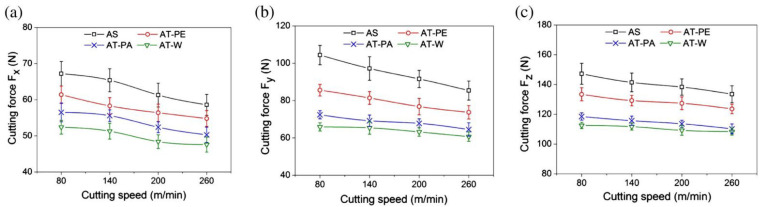
Cutting forces of four kinds of tools at different cutting speeds: (**a**) axial thrust force F_x_; (**b**) radial thrust force F_y_; and (**c**) main force F_z_, (*a_p_* = 0.2 mm, *f* = 0.2 mm/r). AT-PE—linear grooves perpendicular to the cutting edge, AT-PA—linear grooves parallel to the cutting edge, AT-W—and wavy grooves [74].

**Figure 17 materials-15-06945-f017:**
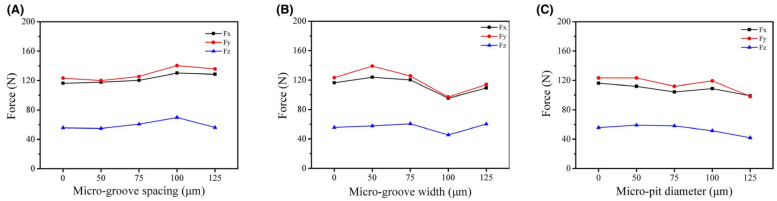
Cutting force of the three tools at the same speed of 120 m/min: (**A**) MG-S; (**B**) MG-W; and (**C**) MP-D [75].

**Figure 18 materials-15-06945-f018:**
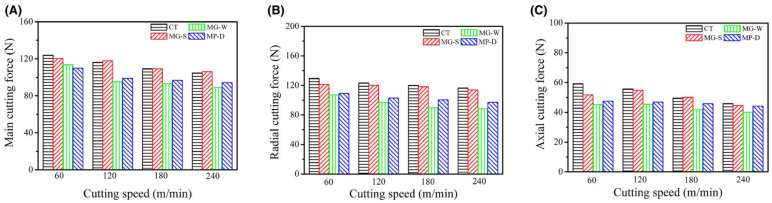
Effect of conventional tools and micro-textured tools on cutting force at speeds varied from 60 to 240 m/min: (**A**) main cutting force; (**B**) radial cutting force; and (**C**) axial cutting force (*a_p_* = 0.2 mm, *f* = 0.102 mm/r) [75].

**Figure 19 materials-15-06945-f019:**
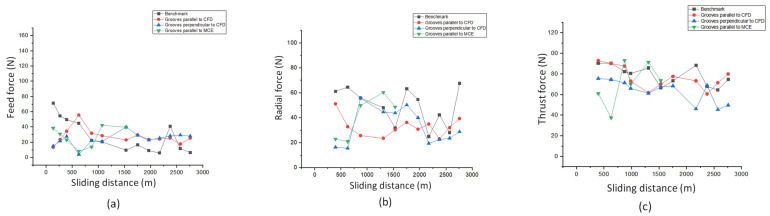
Comparison of the acquired forces while using textured tools and a benchmark cutting tool: (**a**) feed force; (**b**) radial force and; (**c**) thrust force [80].

**Figure 20 materials-15-06945-f020:**
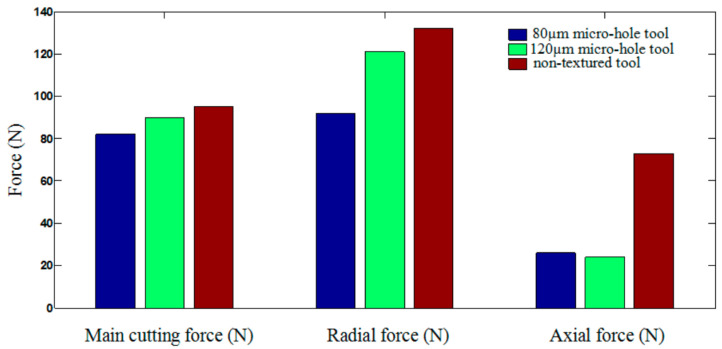
Comparison of cutting forces between micro-hole tool and non-textured tool [78].

**Figure 21 materials-15-06945-f021:**
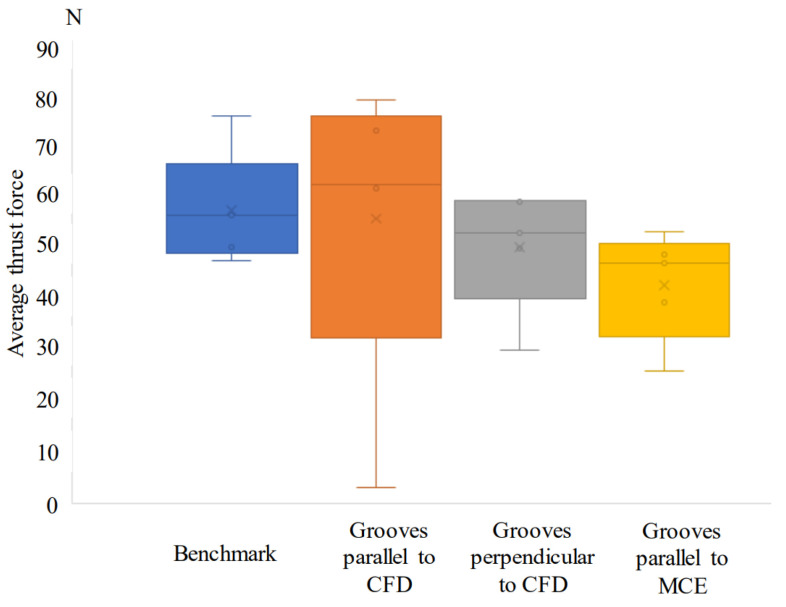
Comparison in average thrust force for benchmark and processed inserts at a cutting length of 873 m [81].

**Figure 22 materials-15-06945-f022:**
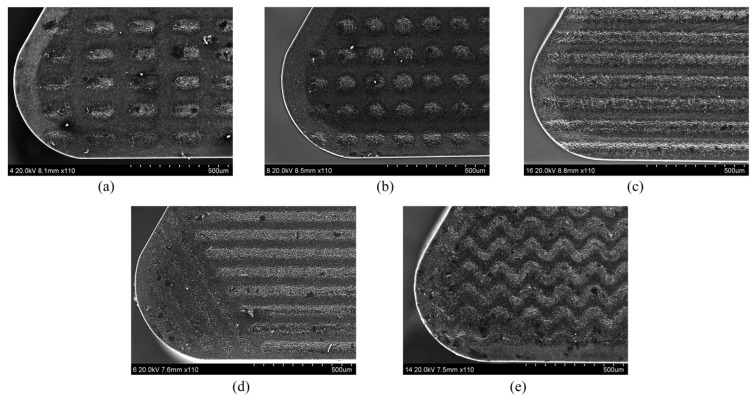
SEM images of five types of micro-textured tools: (**a**) circular pits; (**b**) elliptical grooves; (**c**) transverse grooves; (**d**) composite grooves; (**e**) wavy grooves [82].

**Figure 23 materials-15-06945-f023:**
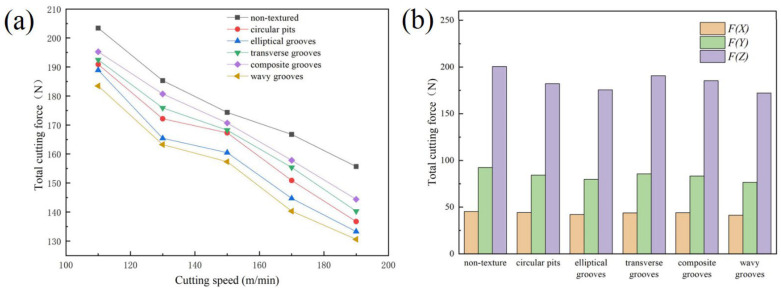
Total cutting force: (**a**) vs. cutting speed predicted by FEA; (**b**) results obtained by cutting experiments [82].

**Figure 24 materials-15-06945-f024:**
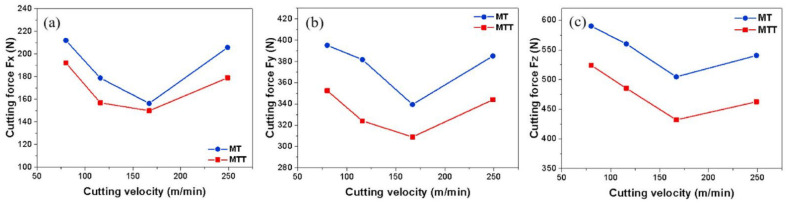
Cutting forces of the MTs and MTTs: (**a**) axial thrust force F_x_; (**b**) radial thrust force F_y_; (**c**) main cutting force F_z_ [76].

**Figure 25 materials-15-06945-f025:**
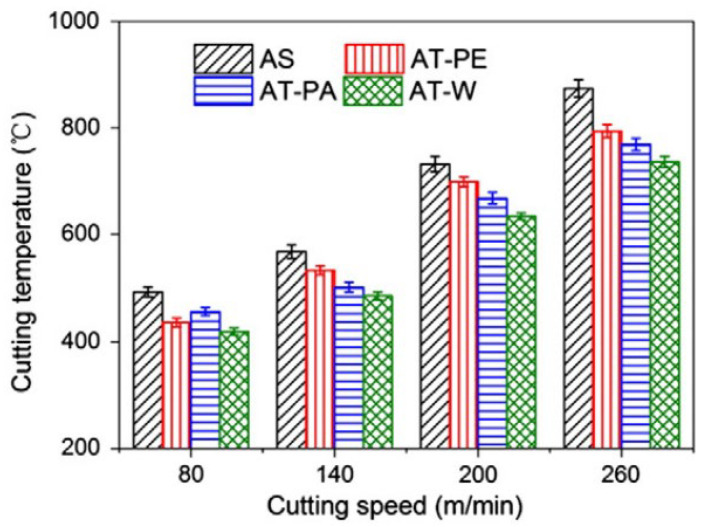
Cutting temperatures at the tool–chip interface of different tools at different cutting speeds (*a_p_* = 0.2 mm, *f* = 0.2 mm/r) [74].

**Figure 26 materials-15-06945-f026:**
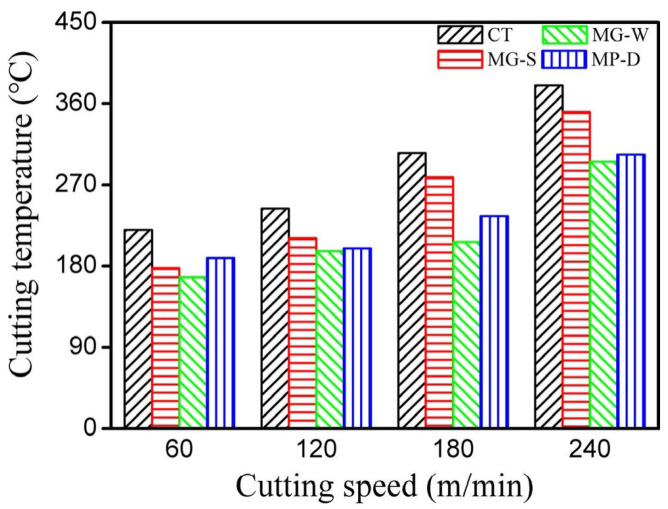
Cutting temperatures at the tool–chip contact interface of different tools at different cutting speeds (*a_p_* = 0.2 mm, *f* = 0.102 mm/r) [75].

**Figure 27 materials-15-06945-f027:**
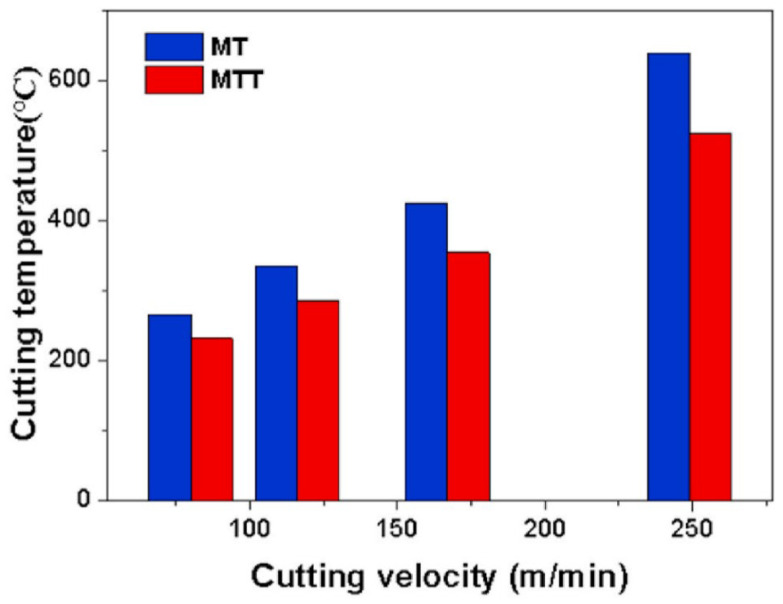
Cutting temperatures of the MT and MTT [76].

**Figure 28 materials-15-06945-f028:**
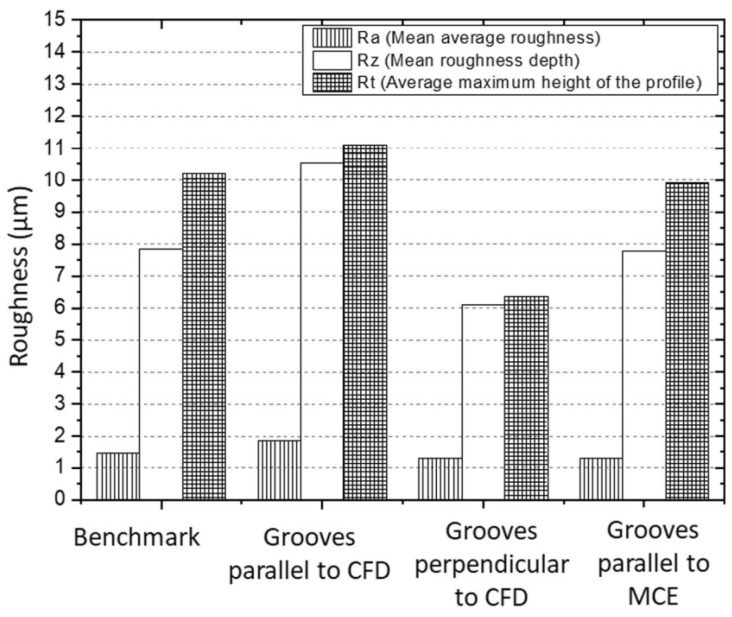
Workpiece surface roughness for benchmark and textured tools [80].

**Figure 29 materials-15-06945-f029:**
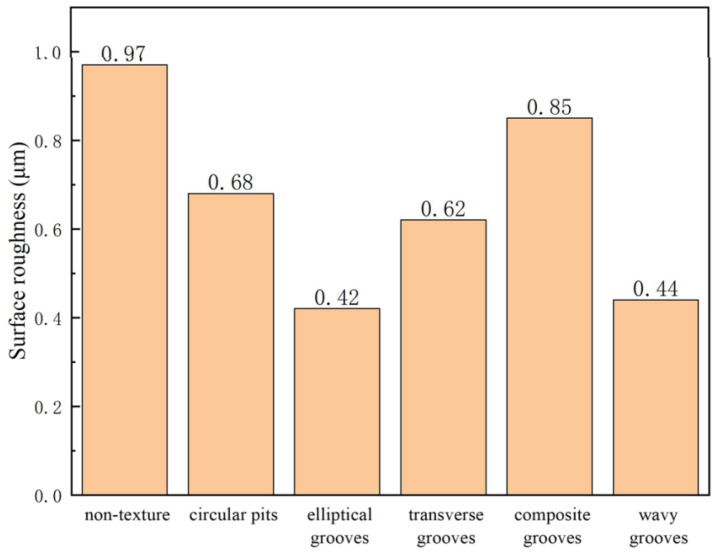
Surface roughness Ra of the machined surfaces obtained by the different tools [82].

**Figure 30 materials-15-06945-f030:**
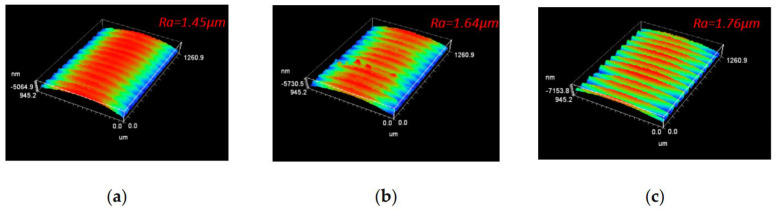
The Surface Roughness Profile of Workpiece: (**a**) d = 80 µm micro-hole tool; (**b**) d = 120 µm micro-hole tool; (**c**) non-textured tool [78].

**Table 1 materials-15-06945-t001:** Paper’s content.

Sections and Subsections	Page Number
1. Introduction	2
2. Methods of creating texturing	3
2.1. Plasma arc machining textures	4
2.2. Laser surface texturing	4
2.3. Electrical discharge machining	5
2.4. Focused ion beam machining	6
2.5. Micro grinding	7
2.6. Conclusions	7
3. Cutting tool surface texturing	8
3.1. Texturing parameters	8
3.2. Most common texture shapes on textured cutting tools	10
3.3. Conclusions	11
4. Effect of surface texturing of ceramic and superhard cutting tools	11
4.1. Effect of surface texturing on friction coefficient, tool wear and adhesive property	12
4.2. Effect of surface texturing on cutting force	21
4.3. Effect of surface texturing on cutting temperature	27
4.4. Effect of surface texturing on machined workpiece roughness	29
4.5. Conclusions	31
5. Conclusions	31
6. References	33
7. Appendix A	
Table A1. Abbreviations, Symbols and Nomenclature	39
Table A2. The most common methods for creating textures on surfaces of different materials and cutting tools	42
Table A3. Classification of several designed textured cutting tools according to the previously proposed categorization	48

**Table 2 materials-15-06945-t002:** Challenges and solutions when using surface texturing in cutting tools.

Challenges	Solutions	Reference
Determining the optimal direction of continuous textures called “groove”.	For cutting tools, most researchers have observed that the use of continuous textures perpendicular to the direction of chip movement shows less wear and a greater reduction in cutting force.	[39]
Decrease in strength of cutting material due to the presence of textures near the cutting edge.	Many researchers have determined that the minimum distance between the cutting edge and the first texture should be three times the feed used.	[84]
Loss of texturing effect due to adhesion of workpiece material in texture cavities that commonly is named “texture blockage”.	The simultaneous implementation of macro and nanotextures leads to the reduction in the workpiece material in the channels of the textures.	[40]
	The use of solid lubricants improves the dry cutting process and reduces the adhesion effect of the machined material.	[28,40]
	The use of multiscale textures inhibiting derivative cutting	[76]

## Data Availability

The data described in this article are openly available in previous works.

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
