# Peer review of "The Influence of Surface Texturing of Ceramic and Superhard Cutting Tools on the Machining Process—A Review"

_materials, 2022, doi:10.3390/ma15196945_

Round 1

Reviewer 1 Report

Overall, the manuscript is composed of good content in terms of research review of surface texturing of ceramic and super hard cutting tools. However, tenses and grammatical issues need to be addressed. A thorough review of sentence and tense structure/ grammar needs to be carried out. 

Author Response

Dear Reviewer 1

Thank you for the professional review and for his/her constructive comment to improve the presentation of materials in our paper. We have considered them to make our paper worthy of publication in the Materials.

Reviewer 1 comment:

Overall, the manuscript is composed of good content in terms of research review of surface texturing of ceramic and super hard cutting tools. However, tenses and grammatical issues need to be addressed. A thorough review of sentence and tense structure/grammar needs to be carried out.

Authors´ answer:

The entire text has been reviewed by a native English speaker and the necessary grammatical corrections have been made.

Reviewer 2 Report

The conducted work “The influence of Surface Texturing of Ceramic and Superhard Cutting Tools on the machining process – A Review” is good. However, following comments should be addressed to further improve paper:

A. GENERAL COMMENTS FOR PAPER ON OVERALL BASIS

1.      Table of content may be added before introduction section.

2.      References mentioned at 75 and 76 needs complete information.

3.      Seven pages and 14 pages tables should not be in main text. Both tables should be reduced and preferably be shown in annexure.

4.      Section 4 is too long. It should have sub headings.

5.      Avoid paragraph of few (2-3) sentences throughout the manuscript, e.g. lines 63-65, etc.

6.      Avoid long sentences throughout the manuscript, e.g. lines 115-119, etc.

B. SPECIFIC COMMENTS FOR IMPROVING FOCUSSED RESEARCH

1.      A table may be added in which the challenges faced by practitioners in current era for frequent cases, its remedies proposed by authors should be mentioned.

2.      At the end of each section, authors opinion and/or comments are welcomed.

Author Response

Dear Reviewer 2

Thank you for the professional review. We have implemented all comments and the resulting changes (apart from the many grammatical adjustments and the errors removed) are marked in blue in the revised manuscript. Below you will find our point by point responses.

We trust the manuscript can now be accepted for publication.

With kind regards, on behalf of all authors

  1. Reviewer 2 comment:

Table of content may be added before introduction section.

Authors´ answer:

A Table of content was added before introduction section.

  1. Reviewer 2 comment:

References mentioned at 75 and 76 needs complete information.

Authors´ answer:

The references 75 and 76 (currently they are references 92 and 93) were completed.

  1. Reviewer 2 comment:

Seven pages and 14 pages tables should not be in main text. Both tables should be reduced and preferably be shown in annexure.

Authors´ answer:

We believe that the information presented in the tables is necessary and reducing it would not have the same impact that the authors hope for future readers. The tables were moved and included in a new section named Appendix.

  1. Reviewer 2 comment:

Section 4 is too long. It should have sub headings.

Authors´ answer:

Section 4 consists of 5 subsections, namely:

4.1. Effect of texturing on friction coefficient, tool wear and adhesive property

4.2. Effect of texturing on cutting force

4.3. Effect of texturing on cutting temperature

4.4. Effect of texturing on cutting roughness

4.5. Conclusions

  1. Reviewer 2 comment:

Avoid paragraph of few (2-3) sentences throughout the manuscript, e.g. lines 63-65, etc.

Authors´ answer:

Paragraphs made up of few sentences have been corrected. For example, in lines 63-65.

  1. Reviewer 2 comment:

Avoid long sentences throughout the manuscript, e.g. lines 115-119, etc.

Authors´ answer:

Every effort has been made to reduce the length of sentences throughout the text. The sentence in lines 115-119 is long because it lists the different machining methods used to obtain textures. Despite this an attempt was made to reduce this sentence as much as possible.

  1. SPECIFIC COMMENTS FOR IMPROVING FOCUSSED RESEARCH

B1. Reviewer 2 comment:

A table may be added in which the challenges faced by practitioners in current era for frequent cases, its remedies proposed by authors should be mentioned.

Authors´ answer:

Table 2 shows the current challenges that mainly take place during the implementation of textures challenges and their Solutions.

B2. Reviewer 2 comment:

At the end of each section, authors opinion and/or comments are welcomed.

Authors´ answer:

At the end of each section a subsection called conclusions has been added, where the summary and opinion of the authors are indicated.

Reviewer 3 Report

The methods for creating textures on the surfaces of different materials and the forms of textures used in cutting tools are two main topics of this paper. Focused on the effect of ceramic and superhard textured cutting tools in improving machining performance of difficult to cut materials. The review of the existing literature reflects the authors’ experience in this field. Before it can be considered for publication, I think some strict and major revisions must be implemented. The detailed comments are presented as follows:

1. The author highlights the effect of ceramic and superhard textured cutting tools in improving machining performance of difficult to cut materials, such as cutting forces and cutting temperature. However, like the roughness of machining surface will also be affected, why did the author not consider.

2. Parts 2 and 3 of this paper summarize the method of tool texture creating and the forms of textures at length, but Part 4 introduces the effect of texture on the cutting performance of ceramic and superhard tools. However, the three parts are not correlated with each other, so the logic needs to be adjusted and condensed.

3. In the relevant content of Chapter 4, when describing the research status or existing problems in a certain aspect, the author should not describe the research content of others' papers in detail, but should briefly summarize the work of others and add some own opinions. And a brief summary should be given at the end of each section.

4. A large number of English abbreviations appear in the paper, and it is recommended that they be explained where they first appear.

5. The scale bar should be added in Figures 11 and 12, and the values in the Figures should remain the same number of bits.

6. The fonts size in the pictures should be in the same format, which is too vague and makes it difficult for the readers to read, especially in Table 1, Figure 3, Figure 13, Figure 23, etc. Please check the full text carefully.

7. Why is the font in italics in lines 167-169 of the text.

8. References (102) and (103) are not cited in the text, and similar problems exist elsewhere in the text. Please check carefully

Author Response

Dear Reviewer 3

Thank you for the professional review. We have implemented all comments and the resulting changes (apart from the many grammatical adjustments and the errors removed) are marked in Red in the revised manuscript. Below you will find our point by point responses.

We trust the manuscript can now be accepted for publication.

With kind regards, on behalf of all authors

Reviewer 3

The methods for creating textures on the surfaces of different materials and the forms of textures used in cutting tools are two main topics of this paper. Focused on the effect of ceramic and superhard textured cutting tools in improving machining performance of difficult to cut materials. The review of the existing literature reflects the authors’ experience in this field. Before it can be considered for publication, I think some strict and major revisions must be implemented. The detailed comments are presented as follows:

  1. Reviewer 3 comment:

The author highlights the effect of ceramic and superhard textured cutting tools in improving machining performance of difficult to cut materials, such as cutting forces and cutting temperature. However, like the roughness of machining surface will also be affected, why did the author not consider.

Authors´ answer:

A new subsection 4.4. named “Effect of surface texturing on cutting roughness” was added to cover the lack of information.

  1. Reviewer 3 comment:

Parts 2 and 3 of this paper summarize the method of tool texture creating and the forms of textures at length, but Part 4 introduces the effect of texture on the cutting performance of ceramic and superhard tools. However, the three parts are not correlated with each other, so the logic needs to be adjusted and condensed.

Authors´ answer:

At the end of each section a subsection called conclusions has been added, where a small explanation that explains the correlation between the adjacent sections

  1. Reviewer 3 comment:

In the relevant content of Chapter 4, when describing the research status or existing problems in a certain aspect, the author should not describe the research content of others' papers in detail, but should briefly summarize the work of others and add some own opinions. And a brief summary should be given at the end of each section.

Authors´ answer:

I thank you very much for your comment. Unfortunately, there is no standard shape of textures which shows a same effect during the machining of different materials at different conditions. The effects of textured cutting tools depend a lot on the machining conditions, the workpiece material, the cutting tool material, its geometry and etc. For this reason, the authors consider that a detailed explanation of each reviewed work is necessary so that future readers can have a clear idea of the research carried out.

  1. Reviewer 3 comment:

A large number of English abbreviations appear in the paper, and it is recommended that they be explained where they first appear.

Authors´ answer:

A table with the abbreviations used throughout the manuscript has been added in the new Appendix section.

  1. Reviewer 3 comment:

The scale bar should be added in Figures 11 and 12, and the values in the Figures should remain the same number of bits.

Authors´ answer:

Figures 11, 12 and 13 are not the result of our work. They were taken from the reference [78], and they have its original quality and presentation. We cannot add any scale bar, because in this case we violate the terms of the permission granted to us to use these images

  1. Reviewer 3 comment:

The fonts size in the pictures should be in the same format, which is too vague and makes it difficult for the readers to read, especially in Table 1, Figure 3, Figure 13, Figure 23, etc. Please check the full text carefully.

Authors´ answer:

All figures in our manuscript are not the result of our work. They were taken from different references, in which the authors used different fonts, and in the present paper the figures show their original quality and presentation. We cannot change the font size in the figure, because in this case we violate the terms of the permission granted to us to use these images

  1. Reviewer 3 comment:

Why is the font in italics in lines 167-169 of the text.

Authors´ answer:

The font in lines 167-169 of the text was corrected

  1. Reviewer 3 comment:

References (102) and (103) are not cited in the text, and similar problems exist elsewhere in the text. Please check carefully

Authors´ answer:

This mistake has been fixed, and references 102 and 103 have been removed.

Round 2

Reviewer 3 Report

The authors have addressed the reviewer comments and the manuscript was improved, so I suggest the paper publishing in Journal. Thanks to authors for their hard work.